# Geometry-Aware Probabilistic Circuits via Voronoi Tessellations

**Sahil Sidheekh** [1]   **Sriraam Natarajan** [1]

## Abstract

Probabilistic circuits (PCs) enable exact and tractable inference but employ data independent mixture weights that limit their ability to capture local geometry of the data manifold. We propose Voronoi tessellations (VT) as a natural way to incorporate geometric structure directly into the sum nodes of a PC. However, naïvely introducing such structure breaks tractability. We formalize this incompatibility and develop two complementary solutions: (1) an approximate inference framework that provides guaranteed lower and upper bounds for inference, and (2) a structural condition for VT under which exact tractable inference is recovered. Finally, we introduce a differentiable relaxation for VT that enables gradient-based learning and empirically validate the resulting approach on standard density estimation tasks.

## 1. Introduction

Probabilistic circuits (PCs) have emerged as a powerful class of generative models that can learn and reason about complex data distributions under uncertainty. By enforcing structural properties, PCs enable exact and linear time inference of likelihoods, marginals and conditionals (Darwiche, 2003; Poon & Domingos, 2011; Choi et al., 2020), making them valuable for applications requiring reliable probabilistic reasoning, such as density estimation, out-of-distribution detection (Braun et al., 2025), causal/counterfactual reasoning (Zečević et al., 2021), multimodal fusion (Sidheekh et al., 2025; 2024) and structured prediction, among others.

Despite recent advances in building expressive PCs, most existing architectures share an important limitation: the mixture weights associated with sum nodes are typically data-independent. This means that the routing decisions within the circuit are fixed globally and do not adapt to individual inputs. While this design choice is often important for preserving tractability, it restricts the ability of PCs to capture and adapt to the local geometric structure in the data manifold. In many real-world distributions, the underlying structure varies across regions of the input space, exhibiting piecewise behavior or locality that cannot be adequately modeled using globally shared mixture weights.

A natural question that arises is: *Can we introduce geometry-aware, input dependent routing into PCs while maintaining tractable inference?* Voronoi tessellations (VT) (Aurenhammer, 1991) offer an appealing geometric mechanism for such routing and have been studied in the context of other generative models such as normalizing flows (Chen et al., 2022). By partitioning the input space into convex polyhedral regions based on proximity to a learned set of centroids, Voronoi cells provide a principled way to assign regions of responsibility to local experts. Incorporating VT within PCs could thus help **achieve geometric interpretability and adaptive routing**, making it a promising approach for modeling distributions with spatially varying structure.

However, naively incorporating Voronoi-based gating into probabilistic circuits creates a fundamental conflict with tractable inference: Voronoi cells are defined by oblique half-space intersections that couple multiple input dimensions. Computing integrals over such convex polyhedron is $\#P$-hard (Dyer & Frieze, 1988) and do not decompose along the variable partitions encoded by the circuit's product nodes. Even when expert distributions are fully factorized, marginalization over Voronoi-gated regions cannot be performed recursively. In deep circuits, these geometric constraints compound across layers, making exact inference intractable even for simple queries.

In this work, we formalize this incompatibility between Voronoi-based routing and tractable inference in PCs, and develop two complementary strategies to address it. First, we present a certified approximate inference framework that preserves the reliability guarantees of PCs by computing provable lower and upper bounds on partition functions, marginals, and conditionals. We achieve this by replacing Voronoi cells with tractable axis-aligned box approximations and propagating bounds through the circuit. Second, we identify a structural condition under which exact tractable inference can be recovered. By factorizing the Voronoi tessellation in a manner that aligns with the circuit

---

[1]The University of Texas at Dallas, Richardson, TX, USA. Correspondence to: Sahil Sidheekh <sahil.sidheekh@utdallas.edu>.

*Proceedings of the 43rd International Conference on Machine Learning*, Seoul, South Korea. PMLR 306, 2026. Copyright 2026 by the author(s).

decomposition, we obtain a class of geometry-aware circuits that support exact inference. This construction, which we refer to as *Hierarchical Factorized Voronoi* (HFV) probabilistic circuits, enforces a shared factorization between the gating mechanism and the expert distributions, thereby restoring recursive integrability. Learning Voronoi tessellated PCs introduces an additional challenge: hard region assignments are typically non-differentiable. To enable end-to-end gradient-based learning, we introduce a soft gating mechanism based on temperature-scaled distance weighting and employ annealing during training. At test time we revert to hard Voronoi assignments, recovering the exact inference guarantees. We prove that this soft-to-hard transition is well-behaved, with exponentially fast convergence as temperature decreases.

Overall, we make the following contributions: (1) We present the first geometric approach based on Voronoi tessellations for training PCs; (2) We formalize the incompatibility between Voronoi-based routing and tractable inference in PCs and outline two different solutions: one based on certified approximate inference and the other based on hierarchical factorizations; (3) For both cases and general geometric modeling, we theoretically analyze the properties of the algorithms and identify the potential gaps; (4) Finally, we perform proof-of-concept experiments to validate the effectiveness and efficiency of the algorithms.

Next, we present the necessary background and position our work in the context of the related work. We then outline the geometric formulation and present the two different strategies for learning. Finally, we present our experiments before concluding by outlining areas of future research.

## 2. Background & Related Work

We begin by introducing the core concepts behind tractable probabilistic models (Poon & Domingos, 2011; Darwiche, 2003; Kisa et al., 2014; Rahman et al., 2014) collectively known as probabilistic circuits(Choi et al., 2020).Although we focus on continuous random variables $\mathbf{X} = \{X_1, \ldots, X_D\}$ with joint domain $\Omega \subseteq \mathbb{R}^D$, the definitions naturally extend to discrete and mixed-variable settings.

**Definition 2.1.** *A **probabilistic circuit** $\mathcal{C}$ over variables $\mathbf{X}$ is a rooted DAG in which each node $n$ is one of the following. A leaf node represents a tractable univariate distribution $p_n(x_i)$ over a single variable $X_i$. A product node computes $f_n(\mathbf{x}) = \prod_{c \in \text{ch}(n)} f_c(\mathbf{x})$. A sum node computes $f_n(\mathbf{x}) = \sum_{c \in \text{ch}(n)} \pi_{n,c} f_c(\mathbf{x})$ with $\pi_{n,c} \geq 0$ and $\sum_c \pi_{n,c} = 1$. The circuit output is $f_{\mathcal{C}}(\mathbf{x}) = f_r(\mathbf{x})$ where $r$ is the root.*

Each node $n$ in a PC is associated with a *scope*, the set of variables it depends on. Sum nodes represent mixtures while product nodes represent factorizations.

**Definition 2.2** (Scope). *The scope of node $n$, denoted*

scope$(n) \subseteq \mathbf{X}$, *is defined recursively. If $n$ is a leaf over $X_i$ then* scope$(n) = \{X_i\}$. *If $n$ is an internal node then* scope$(n) = \bigcup_{c \in \text{ch}(n)}$ scope$(c)$.

Exact tractable inference in a PC hinges on enforcing structural constraints that make marginalization compatible with the circuit factorization. Two key structural properties are:

**Definition 2.3** (Smoothness). *A PC is smooth if for every sum node $n$, all children share the same scope so that* scope$(c) = $ scope$(c')$ *for all $c, c' \in \text{ch}(n)$.*

**Definition 2.4** (Decomposability). *A PC is decomposable if for every product node $n$ with children $c_1, \ldots, c_k$ the scopes are disjoint so that* scope$(c_i) \cap$ scope$(c_j) = \emptyset$ *for all $i \neq j$.*

Smoothness ensures that mixtures are well defined over consistent variable sets, while decomposability ensures that product nodes combine distributions over independent variable sets, which allows integrals to factor. Together, decomposability and smoothness enable efficient exact marginal and conditional inference (Poon & Domingos, 2011; Darwiche, 2003). A third property is:

**Definition 2.5** (Determinism). *A PC is deterministic if for every sum node $(s)$ and input $(\mathbf{x})$ at most one of its child has positive output i.e. $\left| \{ c \in \text{ch}(s) : f_c(\mathbf{x}) > 0 \} \right| \leq 1$.*

Determinism enables efficient MAP inference and yields sparse and interpretable routing behavior.

One of the key research themes within the field of PCs is improving their expressivity to match the performance of deep generative models, while preserving tractability (Sidheekh & Natarajan, 2024; 2026). This has led to tensorized formulations of PCs (Peharz et al., 2020b;a; Liu et al., 2024; Loconte et al., 2025a; Zhang et al., 2025) that can be scaled to millions of parameters and trained efficiently via parallelized computations on GPUs using backpropagation, similar to deep neural networks. However, achieving expressivity through larger circuits or more mixture components often leads to diminishing returns (Liu et al., 2023a). As the number of mixture components grows, optimization becomes harder, parameter redundancy increases, and improvements in likelihood saturate, even though inference remains tractable. To address this, recent works have explored alternative learning paradigms such as latent variable distillation (Liu et al., 2023a;b), where structural or semantic information about the data manifold, extracted using a more expressive teacher model (often a deep generative model), is used as auxiliary supervisory signal to learn a student PC, guiding the latent variables associated with its sum node to be meaningful. Along similar lines, better regularization (Vergari et al., 2015; Ventola et al., 2023; Shih et al., 2021; Dang et al., 2022) and optimization strategies (Zhao et al., 2016; Suresh et al., 2026; Karanam et al., 2025; Liu et al., 2026) have also been proposed to improve generalization.

A complementary direction to increase the expressivity of PCs involves relaxing classical assumptions and extending their representational language. This has resulted in hybrid models that integrate PCs with neural components (Correia et al., 2023; Gala et al., 2024), invertible transformations (Sidheekh et al., 2023), or non-monotonic mixture constructions (Loconte et al., 2024; 2025b). While these models have expanded the representational scope of PCs, they also reveal a recurring challenge: introducing additional dependencies or operations inside the circuit can silently break tractability unless they are carefully aligned with the circuit's factorization structure. For example, in Sidheekh et al. 2023, tractability is recovered by enforcing the invertible neural transformations to satisfy decomposability.

Allowing mixture weights to depend on observed context or on the modeled variables is a natural way to increase model expressivity and enable local specialization, for example, mixture-of-experts (MoE) models (Jacobs et al., 1991; Shazeer et al., 2017), where a learned gating function routes inputs to specialized subnetworks. Within the PC literature, prior approaches have explored similar directions: CSPNs (Shao et al., 2022) parameterize sum-node weights as neural functions of observed features, enabling conditional density estimation over target variables, but sacrificing tractable inference over the conditioning variables. SPQNs (Sharir & Shashua, 2018) introduce quotient nodes to encode conditionals directly, gaining expressive efficiency but restricting tractable marginalization to subsets agreeing with an induced variable ordering (Sharir & Shashua, 2018). Probabilistic neural circuits (PNCs) (Dos Martires, 2024) generalize this idea further by defining the mixing weights as neural functions of ancestor variables, again trading off general tractable marginalization for increased expressivity.

While these approaches are conceptually related, they differ from the setting we study in the source and semantics of the routing signal: they relax or impose constraints on model structure, conditioning, or variable interactions, rather than explicitly encoding geometry in the modeled input space. For instance, CSPNs condition on *external observed features* rather than the modeled variables themselves. Thus, routing is driven by auxiliary information that remains fixed during inference, not on the spatial structure of the data being modeled. SPQNs and PNCs impose *implicit variable orderings* without geometric interpretation, so routing decisions follow graph-theoretic dependencies rather than spatial proximity or geometric regions, and they sacrifice general any order marginalization capabilities of a PC. Similarly, integral circuits and continuous mixtures extend mixture representations using latent variables that are integrated out, rather than inducing explicit geometric partitions of the input space. However, many real-world distributions exhibit strong locality and piecewise structure: different regions of the input space may follow distinct statistical patterns and

dependencies. From a modeling perspective, this suggests routing inputs to local experts based on geometry, rather than relying on globally shared mixture weights.

We posit that such a geometry-aware routing can offer additional benefits beyond likelihood improvement. It can enable interpretability through explicit regions of responsibility, support editability and knowledge incorporation by modifying local components without retraining the entire model, and is naturally suited for online or continual learning scenarios (Veness et al., 2021) where new regions of the space may appear and require minimal adaptation. This motivates the development of geometry-aware PCs, and we aim to build a principled theoretical foundation in this work.

## 3. Geometry-Aware Probabilistic Circuits

A natural way to equip PCs with geometry awareness is to replace the constant (global) sum node weights with *geometry-aware* gating, so that different expert subcircuits specialize to different regions of the data manifold. **Voronoi tessellations** (Aurenhammer, 1991) provide a principled mechanism for such routing. Formally, given centroids $\{\mathbf{c}_1, \ldots, \mathbf{c}_K\} \subset \mathbb{R}^d$, the Voronoi cell of $\mathbf{c}_k$ is defined as

$$V_k = \left\{ \mathbf{u} \in \mathbb{R}^d : \|\mathbf{u} - \mathbf{c}_k\|_2^2 \leq \|\mathbf{u} - \mathbf{c}_j\|_2^2 \ \forall j \right\} \quad (1)$$

Voronoi cells partition space into convex polyhedra with disjoint interiors and boundaries of measure zero, defined by the intersection of half spaces. This assigns inputs to regions based on proximity to learned prototypes, naturally capturing spatial structure, and has been successfully used in clustering (Du et al., 1999), density estimation (Polianskii et al., 2022; Marchetti et al., 2023), and likelihood based generative models (Chen et al., 2022). The resulting routing is *deterministic, interpretable,* and naturally connects to *mixture-of-experts* formulations. Thus, we define a geometry-aware sum node by gating mixture components using a Voronoi partition defined over the node's scope.

**Definition 3.1.** *A **Voronoi-gated sum node** (S) over scope* $\mathbf{X}_S$ *consists of centroids* $\{\mathbf{c}_1, \ldots, \mathbf{c}_K\} \subset \mathbb{R}^{|\mathbf{X}_S|}$ *inducing Voronoi cells* $\{V_k\}$, *mixture weights* $\{\pi_1, \ldots, \pi_K\}$ *with* $\pi_k \geq 0$ *and* $\sum_k \pi_k = 1$, *and child subcircuits* $\{p_1, \ldots, p_K\}$ *each with scope* $\mathbf{X}_S$. *It computes* $f(\mathbf{x}_S) = \sum_{k=1}^K g_k(\mathbf{x}_S) \pi_k p_k(\mathbf{x}_S)$, *where* $g_k(\mathbf{x}_S) = \mathbb{I}[\mathbf{x}_S \in V_k]$

Since the Voronoi cells partition $\mathbb{R}^{|\mathbf{X}_S|}$, exactly one gate is active for almost every $\mathbf{x}_S$, so the node is deterministic up to measure-zero boundaries. However, as we show below, even a single Voronoi-gated sum node can break the factorization of integrals in a PC, making inference intractable.

**Proposition 3.2** (Single-Layer Intractability). *Let* $f(\mathbf{x}) = \sum_{k=1}^K g_k(\mathbf{x}) \pi_k p_k(\mathbf{x})$ *be a Voronoi-gated sum node over* $\mathbf{X} = \{X_1, \ldots, X_D\}$. *Suppose each child is fully factorized,*

$p_k(\mathbf{x}) = \prod_{i=1}^{D} p_k^{(i)}(x_i)$. *Then the partition function $Z = \int f(\mathbf{x})\,d\mathbf{x} = \sum_{k=1}^{K} \pi_k \int_{V_k} p_k(\mathbf{x})\,d\mathbf{x}$ requires integrating $p_k$ over Voronoi cells $V_k$, which are convex polytopes with oblique boundaries. In general, $\int_{V_k} \prod_i p_k^{(i)}(x_i)\,d\mathbf{x}$ does not factor into a product of one-dimensional integrals.*

**Remark 3.3.** *Definition 3.1 places centroids directly in the input space, which corresponds to an identity embedding. We can also use a learned embedding $\phi : \mathbb{R}^{|\mathbf{X}_S|} \to \mathbb{R}^d$ with centroids in $\mathbb{R}^d$ and gates $g_k(\mathbf{x}_S) = \mathbb{I}[\phi(\mathbf{x}_S) \in V_k]$. The negative result shows that the obstruction is the geometry of the gating regions rather than embedding complexity.*

This establishes that the obstruction is already present before considering deeper circuits or more complex expert distributions. Exact tractable inference in smooth decomposable PCs rests on a simple recursion: integrals factor at product nodes because scopes are disjoint, and integrals distribute over sums because mixture weights are constant. Voronoi gating disrupts this recursion by introducing *cell-restricted integrals* of the form $\int_{V_k} p_k(\mathbf{x}_S)\,d\mathbf{x}_S$. Even if $p_k$ factorizes across variables, the region $V_k$ generally does not, as its **oblique facets couple variables that the circuit attempts to separate**. When applied to deep PCs (see appendix), Voronoi gating at multiple scopes induces intersections and projections of such polyhedral constraints across the circuit hierarchy, further preventing the bottom-up factorization needed for exact inference. This is the core incompatibility between geometric routing and circuit factorization. However, geometrically aligning the regions w.r.t a PC's variable decomposition can help retain tractability, as we show next.

**Definition 3.4** (Geometric Alignment). *Consider a partition $\mathbf{X}_S = \mathbf{X}_{S_1} \sqcup \mathbf{X}_{S_2}$. A collection of gating regions $\{R_k\}$ is aligned w.r.t this partition if each region decomposes as $R_k = R_k^{(1)} \times R_k^{(2)}$ with $R_k^{(i)} \subseteq \mathbb{R}^{|\mathbf{X}_{S_i}|}$. Equivalently, membership in $R_k$ can be decided independently as $\mathbf{x}_S \in R_k$ if and only if $\mathbf{x}_{S_1} \in R_k^{(1)}$ and $\mathbf{x}_{S_2} \in R_k^{(2)}$.*

**Theorem 3.5.** *Consider a voronoi gated sum node $f(\mathbf{x}_S) = \sum_k \mathbb{I}[\mathbf{x}_S \in R_k]\ \pi_k\ p_k(\mathbf{x}_S)$ on a variable partition $\mathbf{X}_S = \mathbf{X}_{S_1} \sqcup \mathbf{X}_{S_2}$, where each expert factors as $p_k(\mathbf{x}_S) = p_k^{(1)}(\mathbf{x}_{S_1})\, p_k^{(2)}(\mathbf{x}_{S_2})$. If $\{R_k\}$ is aligned with the partition, then the partition function decomposes as $\int f(\mathbf{x}_S)\,d\mathbf{x}_S = \sum_k \pi_k \left( \int_{R_k^{(1)}} p_k^{(1)}(\mathbf{x}_{S_1})\,d\mathbf{x}_{S_1} \right)\left( \int_{R_k^{(2)}} p_k^{(2)}(\mathbf{x}_{S_2})\,d\mathbf{x}_{S_2} \right).$*

A simple way to satisfy Definition 3.4 is to use axis-aligned partitions, where each $R_k$ is a Cartesian product of intervals (or, more generally, a product of lower-dimensional sets). This includes rectangular boxes, and recovers the tractable region decompositions used implicitly by cutset-style splits (Rahman et al., 2014). However, axis-aligned regions are less expressive than general convex polytopes and often require many rectangles to approximate a single slanted facet. **This motivates two complementary directions**. We can **(1) design geometry-aware gating mechanisms** that

align with the circuit decomposition so that the induced constraints factor compatibly with the circuit structure and **exact tractability is recovered**. Alternatively, we can **(2) accept intractability** to obtain additional expressiveness and derive certified lower and upper bounds on the partition function, which enables **an approximate inference with guarantees**. We now present both approaches.

### 3.1. Certified Approximate Inference

Though geometric gating breaks exact tractability, it is possible retain the reliability of PCs via inference procedures that produce certificates, i.e. provable lower and upper bounds on partition functions, marginals, and conditionals. In this section, we develop a general certified inference framework for Voronoi-Tessellated PCs. The main idea is to replace intractable polyhedral regions with tractable axis-aligned regions for which integration is compatible with decomposability, and to propagate the resulting local bounds through the circuit.

**Bounding Cell-Restricted Integrals with Boxes.** Let $V_k \subset \mathbb{R}^d$ be a Voronoi cell and $\Omega \subseteq \mathbb{R}^d$ a bounded domain. An *inner box $B_k^-$* and *outer box $B_k^+$* satisfy $B_k^- \subseteq V_k \cap \Omega \subseteq B_k^+$, where both are axis-aligned: $B_k^{\pm} = \prod_{i=1}^{d}[a_i^{\pm}, b_i^{\pm}]$. In practice, the domain $\Omega$ may be the full space or a data-dependent bounding box (e.g., per-variable min/max or high-probability truncation). The nesting property yields immediate bounds for any non-negative integrand.

**Lemma 3.6** (Cell Integral Bounds). *Let $p : \mathbb{R}^d \to \mathbb{R}_{\geq 0}$ be a non-negative density and let $B_k^- \subseteq V_k \subseteq B_k^+$. Then $\int_{B_k^-} p(\mathbf{x})\,d\mathbf{x} \leq \int_{V_k} p(\mathbf{x})\,d\mathbf{x} \leq \int_{B_k^+} p(\mathbf{x})\,d\mathbf{x}$*

Thus, the hard part becomes: (i) constructing useful boxes $B_k^{\pm}$, and (ii) integrating circuit outputs over axis-aligned boxes efficiently. We now describe how we construct box approximations for Voronoi cells, optionally restricted to a bounded domain $\Omega = \prod_i [\ell_i, u_i]$.

**Proposition 3.7** (Outer Box Computation). *Let $P_k = V_k \cap \Omega \subseteq \mathbb{R}^d$ where $\Omega = \prod_{i=1}^{d}[\ell_i, u_i]$ is a box domain. The tightest axis-aligned outer box containing $P_k$ is given by $B_k^+ = \prod_{i=1}^{d}\left[ \min_{\mathbf{x} \in P_k} x_i,\ \max_{\mathbf{x} \in P_k} x_i \right]$. Each bound is thus the optimum of a linear program over the polytope $P_k$.*

The LP constraints follow directly from the half-space representation of Voronoi cells together with the box constraints from $\Omega$. We can construct a valid inner box $B_k^-$ by centering it at the centroid $\mathbf{c}_k$ and choosing per-coordinate radii so that the box remains inside all Voronoi half-space constraints. We provide a closed-form conservative construction below.

**Proposition 3.8** (Inner Box Construction). *Assume $\Omega = \mathbb{R}^d$ for simplicity. Let $\delta_k := \min_{j \neq k} \|\mathbf{c}_k - \mathbf{c}_j\|_2$ be the nearest-centroid distance. Then the axis-aligned box $B_k^- = \prod_{i=1}^{d}[\mathbf{c}_{k,i} - r,\ \mathbf{c}_{k,i} + r]$, with $r = \frac{\delta_k}{2\sqrt{d}}$, satisfies $B_k^- \subseteq V_k$.*

**Algorithm 1** Adaptive Anytime Bound Refinement

---

**Require:** Voronoi-PC $\mathcal{C}$, target gap $\epsilon$, max iters $T$, dom. $\Omega$
**Ensure:** Bounds $(Z^-, Z^+)$ with $Z^+ - Z^- \leq \epsilon$ when achievable
1: Initialize partition $\mathcal{P} \leftarrow \{\Omega\}$ and classify all boxes for each cell
2: $(Z^-, Z^+) \leftarrow \text{CERTIFIEDBOUNDS}(\mathcal{C}, \mathcal{P})$
3: **for** $t = 1$ to $T$ **do**
4:     **if** $Z^+ - Z^- \leq \epsilon$ **then**
5:         **return** $(Z^-, Z^+)$
6:     **end if**
7:     // Select boundary box with largest gap contribution
8:     $(n^*, k^*, B^*) \leftarrow \arg\max$ gap contribution over BOUNDARY boxes
9:     // where $w_{n,k}$ is weight in global bound computation
10:     $j^* \leftarrow \arg\max_j (b_j - a_j)$ for $B^* = \prod_j [a_j, b_j]$
        // Longest dimension
11:    Bisect $B^*$ along dimension $j^*$ into $B_L, B_R$
12:    Remove $B^*$ from $\mathcal{P}$; add $B_L, B_R$ to $\mathcal{P}$
13:    Reclassify $B_L, B_R$ for all cells using memb. tests
14:    $(Z^-, Z^+) \leftarrow \text{CERTIFIEDBOUNDS}(\mathcal{C}, \mathcal{P})$
15: **end for**
16: **return** $(Z^-, Z^+)$

---

**Algorithm 2** Certified Bound Computation

---

**Require:** Voronoi-gated PC $\mathcal{C}$, box approximations $\{(B_k^-, B_k^+)\}$ for each Voronoi cell
**Ensure:** Bounds $(Z^-, Z^+)$ on $Z = \int f_{\mathcal{C}}(\mathbf{x})\, d\mathbf{x}$
1: **for** each node $n$ in reverse topological order **do**
2:     **if** $n$ is a leaf **then**
3:         $I_n^- \leftarrow \int p_n$;   $I_n^+ \leftarrow \int p_n$
4:     **else if** $n$ is a product node **then**
5:         $I_n^- \leftarrow \prod_{c \in \text{ch}(n)} I_c^-$;   $I_n^+ \leftarrow \prod_{c \in \text{ch}(n)} I_c^+$
6:     **else if** $n$ is a standard sum node **then**
7:         $I_n^- \leftarrow \sum_c \pi_{n,c} I_c^-$;   $I_n^+ \leftarrow \sum_c \pi_{n,c} I_c^+$
8:     **else if** $n$ is a Voronoi-gated sum node **then**
9:         **for** each cell $k$ **do**
10:            $J_k^- \leftarrow \text{INTEGRATEBOX}(p_k, B_k^-)$
11:            $J_k^+ \leftarrow \text{INTEGRATEBOX}(p_k, B_k^+)$
12:        **end for**
13:        $I_n^- \leftarrow \sum_k \pi_k J_k^-$;   $I_n^+ \leftarrow \sum_k \pi_k J_k^+$
14:    **end if**
15: **end for**
16: **return** $(I_{\text{root}}^-, I_{\text{root}}^+)$

---

When $\Omega$ is bounded, we can simply intersect $B_k^-$ with $\Omega$ to preserve containment within $V_k \cap \Omega$: $B_k^- \leftarrow B_k^- \cap \Omega$. This construction trades tightness for simplicity and robustness. If tighter inner boxes are needed we can optimize radii $r_i$ per dimension subject to the Voronoi half-space constraints.

### 3.1.1. ANYTIME BOUND REFINEMENT

The basic box approximations above provide valid certified bounds but may be loose, particularly in high dimensions. We thus develop an anytime refinement algorithm that monotonically tightens the bounds through recursive box subdivision, enabling flexible trade-offs between computational cost and bound quality. The core idea is to recursively bisect boxes and test sub-boxes for containment or intersection with Voronoi cells. For outer boxes, we can discard sub-boxes that don't intersect the cell, and for inner boxes, we retain sub-boxes fully contained within the cell. Formally, we maintain a disjoint axis-aligned partition $\mathcal{P}$ of domain $\Omega_S = \prod_{i \in S} [\ell_i, u_i]$ where each box $B \in \mathcal{P}$ is labeled for each Voronoi cell $V_k$ as: INSIDE if $B \subseteq V_k$, OUTSIDE if $B \cap V_k = \emptyset$, or BOUNDARY otherwise. This induces approximations $V_k^-(\mathcal{P}) := \bigcup_{B:\, \text{lab}_k(B)=\text{INSIDE}} B$ and $V_k^+(\mathcal{P}) := \bigcup_{B:\, \text{lab}_k(B) \neq \text{OUTSIDE}} B$ satisfying $V_k^-(\mathcal{P}) \subseteq V_k \cap \Omega_S \subseteq V_k^+(\mathcal{P})$. Since $\mathcal{P}$ is disjoint, integration decomposes additively with node bounds $\sum_k \pi_k I_k^{\pm}(\mathcal{P})$ propagating via Theorem 3.10. Classification can be done using the half-space representation $V_k = \bigcap_{j \neq k} \{\mathbf{x} : (\mathbf{c}_j - \mathbf{c}_k)^\top \mathbf{x} \leq$

$\frac{1}{2}(\|\mathbf{c}_j\|^2 - \|\mathbf{c}_k\|^2)\}$. For box $B = \prod_i [l_i, u_i]$ and half-space normal $a$, the extrema $\max_{\mathbf{x} \in B} a^\top \mathbf{x}$ and $\min_{\mathbf{x} \in B} a^\top \mathbf{x}$ are computed by selecting $u_i$ (resp. $l_i$) when $a_i \geq 0$ for the maximum, and vice versa for the minimum. Then $B \subseteq V_k$ iff all half-space maxima satisfy the constraint, and $B \cap V_k = \emptyset$ if any half-space minimum violates its constraint. Refinement bisects BOUNDARY boxes at midpoints, replaces them with disjoint children, reclassifies, and recomputes bounds, prioritizing boxes with largest gap contribution. Algorithm 1 outlines the refinement, and the below theorem establishes its monotone tightening and convergence properties.

**Theorem 3.9.** *Let $\mathcal{P}_t$ denote the partition after $t$ refinement steps with bounds $(Z_t^-, Z_t^+)$. Then (i) $Z_t^- \leq Z \leq Z_t^+ \,\forall\, t$ (ii) $Z_t^- \leq Z_{t+1}^-$ and $Z_{t+1}^+ \leq Z_t^+$ for all $t$ (iii) if refinement drives the boundary volume $\mu(V_k^+(\mathcal{P}_t) \setminus V_k^-(\mathcal{P}_t)) \to 0$ for each cell, then $\lim_{t \to \infty} Z_t^\pm = Z$. Under uniform refinement, the gap scales roughly as $Z_t^+ - Z_t^- = O(2^{-t/d})(Z_0^+ - Z_0^-)$, requiring depth $O(d \log(1/\epsilon))$ to achieve target gap $\epsilon$.*

### 3.1.2. PROPAGATING BOUNDS THROUGH THE CIRCUIT.

We next show how to propagate local cell bounds through the sum and product structure of the circuit. We first focus on the partition function $(Z)$, and later discuss how the same machinery applies to marginals and conditionals.

**Theorem 3.10** (Bound Propagation). *Let $\mathcal{C}$ be a Voronoi-gated PC. For each node $n$ let $I_n = \int f_n(\mathbf{x})\, d\mathbf{x}$ denote the integral of the subcircuit rooted at $n$, and let $(I_n^-, I_n^+)$ denote certified bounds on $I_n$. We can compute bounds bottom up as follows. If $n$ is a leaf, then $I_n^- = I_n^+ = \int p_n(x)\, dx$. If $n$ is a product node with children $\{c_j\}$ and disjoint scopes, then $I_n^- = \prod_j I_{c_j}^-$,    $I_n^+ = \prod_j I_{c_j}^+$. If $n$ is a*

*standard sum node with children* $\{c_j\}$ *and weights* $\{\pi_{n,c_j}\}$, *then* $I_n^- = \sum_j \pi_{n,c_j} I_{c_j}^-$, $I_n^+ = \sum_j \pi_{n,c_j} I_{c_j}^+$. *If* $n$ *is a Voronoi-gated sum node with children* $\{p_k\}$ *and cell boxes* $\{(B_k^-, B_k^+)\}$, *then* $I_n^- = \sum_k \pi_k \int_{B_k^-} p_k(\mathbf{x}) \, d\mathbf{x}$; $I_n^+ = \sum_k \pi_k \int_{B_k^+} p_k(\mathbf{x}) \, d\mathbf{x}$. *At the root, we obtain bounds* $(Z^-, Z^+)$ *satisfying* $Z^- \le Z \le Z^+$.

Algorithm 2 summarizes the bound computation. The only non-standard operation is integrating a decomposable sub-circuit over a box, which as we show in Theorem 3.14 can be implemented recursively by applying interval integration at leaves and factorization at product nodes. We can apply the same machinery to bound marginals and conditionals.

**Corollary 3.11** (Marginal Bounds). *For* $p(\mathbf{x}_A) = \int p(\mathbf{x}) \, d\mathbf{x}_{\bar{A}}$ *we obtain bounds* $(p^-(\mathbf{x}_A), p^+(\mathbf{x}_A))$ *by propagating box restricted bounds through the circuit after restricting each box approximation to the* $\bar{A}$ *dimensions.*

**Corollary 3.12** (Conditional Bounds). *For disjoint* $A, B$ *and* $p(\mathbf{x}_A \mid \mathbf{x}_B) = p(\mathbf{x}_A, \mathbf{x}_B) / p(\mathbf{x}_B)$, *if we have bounds* $p^-(\mathbf{x}_A, \mathbf{x}_B) \le p(\mathbf{x}_A, \mathbf{x}_B) \le p^+(\mathbf{x}_A, \mathbf{x}_B)$ *and* $p^-(\mathbf{x}_B) \le p(\mathbf{x}_B) \le p^+(\mathbf{x}_B)$ *with* $p^-(\mathbf{x}_B) > 0$, *then* $\frac{p^-(\mathbf{x}_A, \mathbf{x}_B)}{p^+(\mathbf{x}_B)} \le p(\mathbf{x}_A \mid \mathbf{x}_B) \le \frac{p^+(\mathbf{x}_A, \mathbf{x}_B)}{p^-(\mathbf{x}_B)}$.

Certified bounds provide guaranteed inference in settings where exact computation is intractable, but they have clear limitations. In high dimensions the initial inner boxes may capture only a small fraction of each Voronoi cell, leading to loose initial bounds. Tight bounds require many refinements, and the refinement cost grows quickly with dimension and the number of cells. Finally, while refinement can approximate exact inference arbitrarily well, it does not restore exactness at finite computation. This motivates designing Voronoi gating mechanisms that align with circuit decomposition and recover exact tractable inference.

### 3.2. Hierarchical Factorized Voronoi PCs

Next, we show how to enforce geometric alignment by design through *hierarchical factorization* of Voronoi tessellations to retain tractable inference. The key idea is to partition the Voronoi centroids in a manner that mirrors the circuit's vtree structure, ensuring that at each product node, the induced Voronoi cells decompose into independent factors over disjoint variable subsets. Let a scope $\mathbf{X}_S$ be partitioned into disjoint blocks $\mathbf{X}_S = \bigsqcup_{i=1}^m \mathbf{X}_{S_i}$. For each block $i$, choose centroids $\{\mathbf{c}_1^{(i)}, \ldots, \mathbf{c}_{K_i}^{(i)}\} \subset \mathbb{R}^{|\mathbf{X}_{S_i}|}$ and let $\{V_{k_i}^{(i)}\}_{k_i=1}^{K_i}$ be the induced Voronoi cells in $\mathbb{R}^{|\mathbf{X}_{S_i}|}$. These induce a *joint* partition of $\mathbb{R}^{|\mathbf{X}_S|}$ into product cells indexed by $\mathbf{k} = (k_1, \ldots, k_m)$ as: $V_{\mathbf{k}} = V_{k_1}^{(1)} \times \cdots \times V_{k_m}^{(m)}$. The corresponding hard gate factors along blocks: $g_{\mathbf{k}}(\mathbf{x}_S) = \mathbb{I}[\mathbf{x}_S \in V_{\mathbf{k}}] = \prod_{i=1}^m \mathbb{I}[\mathbf{x}_{S_i} \in V_{k_i}^{(i)}] = \prod_{i=1}^m g_{k_i}^{(i)}(\mathbf{x}_{S_i})$, so membership can be decided independently within each

block. The defining feature is that each joint cell is a Cartesian product of lower-dimensional Voronoi cells, which is exactly the geometric analogue of decomposability. We can now define a gated sum node that uses a factorized Voronoi partition and couples it to a matching factorized expert.

**Definition 3.13** (HFV-Gated Sum Node). *Let* $\mathbf{X}_S = \bigsqcup_{i=1}^m \mathbf{X}_{S_i}$. *An HFV-gated sum node over scope* $\mathbf{X}_S$ *consists of a factorized Voronoi partition with* $K_i$ *cells on factor* $i$, *mixture weights* $\{\pi_{\mathbf{k}}\}_{\mathbf{k} \in [K_1] \times \cdots \times [K_m]}$ *with* $\sum_{\mathbf{k}} \pi_{\mathbf{k}} = 1$, *and factor subcircuits* $\{p_{k_i}^{(i)}\}$ *where* $p_{k_i}^{(i)}$ *has scope* $\mathbf{X}_{S_i}$. *The node computes* $f(\mathbf{x}_S) = \sum_{\mathbf{k}} g_{\mathbf{k}}(\mathbf{x}_S) \pi_{\mathbf{k}} \prod_{i=1}^m p_{k_i}^{(i)}(\mathbf{x}_{S_i})$, *where* $g_{\mathbf{k}}(\mathbf{x}_S) = \prod_{i=1}^m g_{k_i}^{(i)}(\mathbf{x}_{S_i})$.

The critical design choice is that the gate and the expert share the same factorization pattern. This alignment restores the ability to apply Fubini's theorem [1] to reduce high-dimensional integrals into products of lower-dimensional integrals. To obtain a full circuit over $\mathbf{X} = \{X_1, \ldots, X_D\}$, we align HFV gating with a variable tree (vtree), so that at each internal vtree node the gate factors across its left/right child scopes. This yields a hierarchical (multi-resolution) geometric partition: coarse routing happens at higher scopes, while finer routing refines decisions within smaller scopes. We now show that HFV restores exact tractable inference. The result mirrors standard PC tractability, but incurs an additional multiplicative factor that reflects the number of joint Voronoi cell combinations at each gated sum.

**Theorem 3.14** (Tractability of HFV-PCs). *Let* $\mathcal{C}$ *be an HFV-PC with* $|\mathcal{C}|$ *nodes, maximum factorization degree* $m$, *and at most* $K$ *Voronoi cells per factor. Then the partition function, marginals, and conditionals are computable exactly in time* $O(|\mathcal{C}| K^m)$. *For binary HFV-PCs the time is* $O(|\mathcal{C}| K^2)$.

HFV intentionally restricts geometry to preserve decomposability. By requiring each joint cell to be a Cartesian product of lower-dimensional Voronoi cells, HFV can represent rich piecewise structure *within* each scope block and refine it hierarchically, but cannot realize arbitrary oblique polytopes whose half-space normals couple variables *across* different scope blocks. A concrete example is a dataset separated by a rotated hyperplane such as $x_1 + x_2 = c$: VT can represent such a boundary directly with a single centroid pair, whereas HFV must approximate it through a hierarchical composition of factorized axis-aligned regions. The two constructions therefore occupy different points in the expressivity-tractability tradeoff. **HFV-PC** is preferable when exact inference is **required** and the relevant geometric structure factorizes along the circuit decomposition, while **VT-PC** is preferable when cross-variable boundaries are essential and one is willing to accept certified approximate inference for added expressivity. In higher dimensions, HFV gating can also be applied selectively to a subset of PC lay-

---

[1] Proofs are provided in the appendix

ers, avoiding full determinism while introducing geometry awareness in the gated layers.

### 3.3. Learning via Soft Gating

Both geometry-aware constructions we have introduced in this paper rely on hard routing, where inputs activate a single region. This yields crisp locality and interpretability and enables exact inference in HFV-PCs as well as certified bounds for general VT-PCs. However, hard Voronoi assignments are non-differentiable, preventing gradient-based learning of the centroids together with the PC parameters. We thus introduce a smooth relaxation that supports standard backpropagation during training, and revert to hard gating at test time to recover the desired inference guarantees.

**Definition 3.15** (Soft Voronoi Gate). *Given centroids* $\{\mathbf{c}_1, \ldots, \mathbf{c}_K\} \subset \mathbb{R}^d$ *and an inverse temperature* $\alpha > 0$, *we define the soft Voronoi gate*

$$w_k(\mathbf{u}; \alpha) = \frac{\exp(-\alpha \|\mathbf{u} - \mathbf{c}_k\|^2)}{\sum_{j=1}^K \exp(-\alpha \|\mathbf{u} - \mathbf{c}_j\|^2)}. \qquad (2)$$

*For any* $\mathbf{u}$, *the weights satisfy* $w_k(\mathbf{u}; \alpha) > 0$ *and* $\sum_{k=1}^K w_k(\mathbf{u}; \alpha) = 1$ *and are smooth in both* $\mathbf{u}$ *and* $\{\mathbf{c}_k\}$

The temperature parameter controls the sharpness of routing. When $\alpha$ is small the weights are diffuse, and when $\alpha$ is large the weights concentrate on the nearest centroid. Correspondingly, a soft Voronoi-gated sum node computes: $f(\mathbf{x}_S; \alpha) = \sum_{k=1}^K w_k(\mathbf{x}_S; \alpha) \pi_k p_k(\mathbf{x}_S)$. For HFV-PCs, factorization can be maintained by applying soft gates per factor. Given partition $\mathbf{X}_S = \bigsqcup_{i=1}^m \mathbf{X}_{S_i}$, define a factorized soft gate $w_{\mathbf{k}}(\mathbf{x}_S; \alpha) = \prod_{i=1}^m w_{k_i}^{(i)}(\mathbf{x}_{S_i}; \alpha)$ where each factor uses centroids in $\mathbb{R}^{|\mathbf{X}_{S_i}|}$. This ensures gradient signals to factor-$i$ centroids depend only on variables in $\mathbf{X}_{S_i}$, preserving the decomposition structure during optimization. For any $\alpha > 0$, the soft gates are smooth and the circuit likelihood is differentiable with respect to all parameters. The centroid gradient has the form $\nabla_{\mathbf{c}_k} w_k(\mathbf{u}; \alpha) = 2\alpha w_k(\mathbf{u}; \alpha)(1 - w_k(\mathbf{u}; \alpha))(\mathbf{u} - \mathbf{c}_k)$ The factor $w_k(1 - w_k)$ peaks when the gate is uncertain (near decision boundaries), and vanishes when routing is already confident ($w_k \approx 0$ or $1$). Thus centroids are primarily updated in regions where the current tessellation is contested.

**Soft-to-Hard Convergence.** We train VT-PCs and HFV-PCs via soft gates but ultimately require the exact inference guarantees of hard HFV gating. The next result highlights that increasing $\alpha$ recovers hard Voronoi assignments, with an exponential rate governed by a geometric margin.

**Theorem 3.16** (Soft-to-Hard Convergence). *Let* $g_k(\mathbf{u}) = \mathbb{I}[\mathbf{u} \in V_k]$ *be the hard Voronoi gate induced by* $\{\mathbf{c}_k\}$ *and let* $w_k(\mathbf{u}; \alpha)$ *be the soft gate in* (2). *Then for any* $\mathbf{u}$ *not lying on a Voronoi boundary,* $\lim_{\alpha \to \infty} w_k(\mathbf{u}; \alpha) = g_k(\mathbf{u})$.

*Moreover, if* $k^*(\mathbf{u}) = \arg \min_k \|\mathbf{u} - \mathbf{c}_k\|$ *and the margin* $\gamma(\mathbf{u}) = \min_{j \neq k^*(\mathbf{u})} (\|\mathbf{u} - \mathbf{c}_j\|^2 - \|\mathbf{u} - \mathbf{c}_{k^*}\|^2)$ *is positive, then* $1 - w_{k^*}(\mathbf{u}; \alpha) \leq (K - 1)e^{-\alpha \gamma(\mathbf{u})}$. *Further, if* $p$ *is integrable, then* $w_k(\cdot; \alpha) \to g_k(\cdot)$ *also implies* $\lim_{\alpha \to \infty} \int w_k(\mathbf{u}; \alpha) p(\mathbf{u}) \, d\mathbf{u} = \int_{V_k} p(\mathbf{u}) \, d\mathbf{u}$.

The gap $\gamma(\mathbf{u})$ quantifies how much closer $\mathbf{u}$ is to its nearest centroid than to the runner-up. Larger margins mean the softmax ratios $\exp(-\alpha(d_j - d_{k^*}))$ decay faster, so routing becomes effectively hard at smaller $\alpha$. The exponential bound formalizes this geometric picture and justifies annealing: as training progresses, increasing $\alpha$ sharpens routing while remaining stable on points that are already well-separated by the current centroids.

**Training.** We thus train soft HFV-PCs and VT-PCs via maximum-likelihood, gradually increasing $\alpha$ so that routing sharpens over time. This annealing schedule avoids early training instabilities where centroids collapse or assignments become overly brittle before the experts have adapted. We use softmax projection to enforce $\pi_{\mathbf{k}} \geq 0$ and $\sum_{\mathbf{k}} \pi_{\mathbf{k}} = 1$. After training we use the learned centroids to define a Voronoi tessellation and perform inference in one of two modes. Hard-gated inference, which replaces $w_{k_i}^{(i)}$ with $g_{k_i}^{(i)}$ and recovers the exact tractability guarantees. This is the default mode we use in experiments because it preserves exact partition functions and exact marginals. Soft-gated inference uses a finite $\alpha$. This yields a fully smooth model, but exact marginalization generally requires integrating smooth gate functions, which typically forfeit the exact HFV guarantees. We therefore treat soft gating primarily as a training device and hard gating as the inference-time model.

## 4. Experiments & Results

### 4.1. Low Dimensional Geometric Datasets

To validate our theoretical framework, we first consider eight synthetic distributions where geometric structure is explicit and exact verification is feasible: four $2D$ (Alphabet, CheckerBoard, Pinwheel, Spiral) and four $3D$ (BentLissajous, InterlockedCircles, Knotted, TwistedEight) (Sidheekh et al., 2022; 2023), each with 10k train, 5k validation, and 5k test samples. We compare two notable base PC architectures: EinsumNet (random binary tree region graph) (Peharz et al., 2020a) and HCLTs (Hidden Chow-Liu trees) (Liu & Van den Broeck, 2021) against their geometry-aware variants: VT-EinsumNet/HCLT using Voronoi tessellations at root sum nodes with certified approximate inference, and HFV-EinsumNet/HCLT using hierarchical factorized Voronoi gating aligned with circuit decomposition for exact inference. We use Gaussian leaves and Tucker sum-product layers in all models (Loconte et al., 2025a), keeping the base architecture same, with 10 input and sum units for $3D$ (and

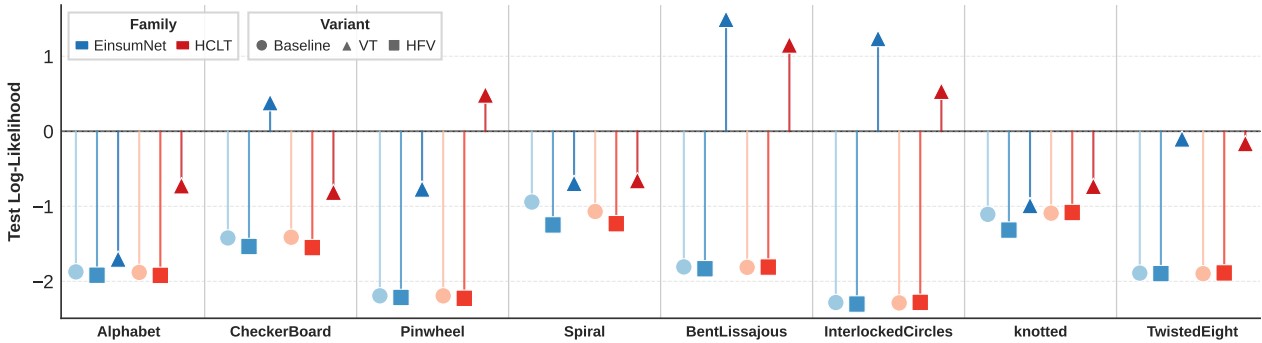

*Figure 1.* **Mean Test Log-likelihood** (↑) on **synthetic** 2D and 3D density estimation tasks achieved by EinsumNet and HCLT along with their geometry-aware extensions using Voronoi tessellations (VT) and hierarchical factorized Voronoi (HFV), averaged across 3 trials. For VT, values correspond to the lower bound on the log-likelihood obtained via our certified approximate inference framework.

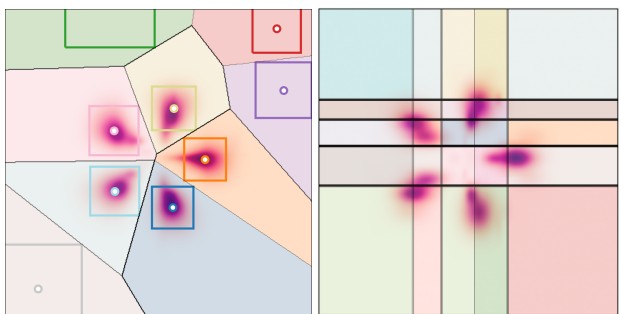

*Figure 2.* Visualization of the distribution and voronoi tessellations learned by a VT-EinsumNet (left) and HFV-EinsumNet (right) on the 2D pinwheel dataset. The axis aligned boxes in the left figure represent the Inner Boxes computed for estimating the lower bound on the partition function using our conservative construction.

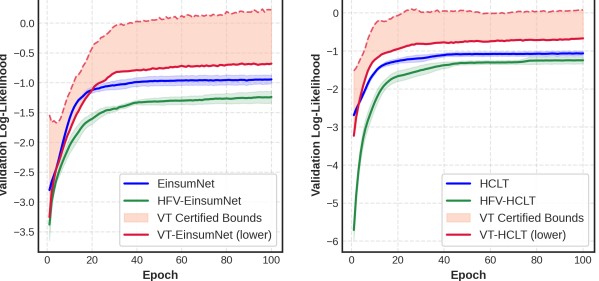

*Figure 3.* **Learning curves on the 2D spiral dataset.** Validation log-likelihood across epochs for EinsumNet (left) and HCLT (right), averaged over 3 trials. VT models report a *certified* lower bound while baselines and HFV use exact inference.

5 each for 2D) datasets, for a fair comparison. We employ maximum likelihood via stochastic gradient descent to learn the parameters of the PC and VT, by training using an Adam optimizer with a learning rate of 0.01 and batch size of 500 for 100 epochs. For VT and HFV variants, we employ soft gating with linear temperature annealing for and k-means centroid initialization (100 iterations), and switch to hard gating at test time. VT models report certified lower bounds using inner box approximations without refinement.

**Results.** Figure 1 depicts the mean test log-likelihoods across all datasets and model variants. VT models (triangles) consistently achieve strong performance, with their certified lower bounds often exceeding baseline exact log-likelihoods, demonstrating that the increased expressivity from unconstrained geometry-aware routing captures structure missed by input-independent weights, even when accounting for conservative approximation gaps. HFV models (squares) achieve performance comparable to baselines (circles). This is expected in our low-dimensional, shallow-circuit regime, as the alignment constraints required for exact tractability in HFV induce fully deterministic, factorized partitions,

which can reduce expressive power relative to smooth decomposable PCs. The advantage of **HFV here however, is conceptual and algorithmic**: it retains *exact tractable inference while providing an explicit geometric interpretation* that can be useful for downstream tasks such as continual learning. Figure 2 visualizes the learned routing structure on the 2D pinwheel dataset. In VT (left), the Voronoi cells adapt to the arms of the distribution and assign regions of responsibility to local experts; the axis-aligned *inner boxes* shown are the conservative subsets used by our certified lower-bound computation, and we see that in practice they capture the majority of the modeled probability mass within each cell on this dataset. In HFV (right), the partition is hierarchical and factorized, yielding axis-aligned regions that preserve exact tractability by construction. Together, these overlays offer **an interpretable view of *where* specialization occurs** and help explain why geometry-aware routing is effective on distributions with strong locality.

Figure 3 shows validation learning curves of EinsumNet and HCLT on the 2D spiral dataset. For VT we plot the certified **lower bound** (solid red) together with the **certified interval** induced by the partition function $Z \in [Z^-, Z^+]$

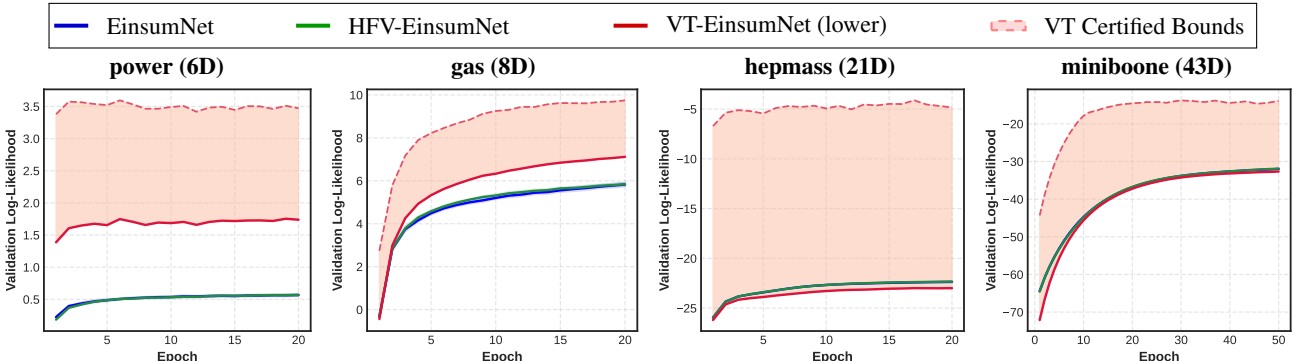

*Figure 4.* **Learning curves on the UCI tabular datasets.** Validation log-likelihood across epochs on UCI tabular benchmarks (EinsumNet family), ordered by increasing dimensionality. VT models report a *certified* lower bound while baselines and HFV use exact inference.

*Table 1.* Mean test log-likelihood (↑) on UCI tabular benchmarks, averaged over 3 trials. VT rows report certified lower [LO] and upper [UP] bounds without additional refinement.

| Model | power (6D) | gas (8D) | hepmass (21D) | miniboone (43D) |
|---|---|---|---|---|
| EinsumNet | 0.56 | 5.85 | −22.35 | −32.24 |
| HFV-EinsumNet | 0.56 | 5.86 | **−22.32** | **−32.18** |
| VT-EinsumNet [LO] | **1.73** | **7.13** | −22.98 | −32.82 |
| VT-EinsumNet [UP] | 3.46 | 9.76 | −4.82 | −14.07 |
| HCLT | 0.55 | 7.22 | −20.36 | −25.43 |
| HFV-HCLT | 0.55 | 7.07 | −20.35 | **−25.36** |
| VT-HCLT [LO] | **5.10** | **10.57** | **−19.10** | −26.46 |
| VT-HCLT [UP] | 6.77 | 13.17 | −1.61 | −8.95 |

Table 1 reports mean test log-likelihoods averaged over 3 trials. VT certified lower bounds exceed or closely match the exact likelihoods of corresponding baselines on the majority of datasets. VT-HCLT improves over the HCLT-baselines on **power**, **gas**, and **hepmass**, and VT-EinsumNet improves over its baseline on **power** and **gas**. But since these are lower bounds, the true VT likelihoods can be expected to be higher still. In higher-dimensions, we also observe that the gap between lower and upper bounds widens. This is expected from the conservative inner/outer box construction and is consistent with the dimension dependence suggested by Theorem 3.9. Importantly, even the lower bound remains competitive or superior, confirming that the framework extends meaningfully beyond the 2D/3D synthetic settings.

## 5. Conclusion

We developed a principled foundation for introducing geometric awareness into PCs, by replacing constant sum-node weights with geometry driven gates induced via Voronoi tessellations. Our key message is that *retaining tractable inference requires geometric alignment with the circuits factorization*, as general Voronoi partitioning breaks recursive marginalization. We addressed this tension in two ways: first, we developed a theoretically grounded certified approximate inference framework that preserved expressivity while providing reliable inference. Second, we identified structural properties on the tessellation under which tractable inference was recovered. Finally, we presented a soft relaxation with convergence guarantees that enabled stable gradient based learning, and empirically validated the resulting model on synthetic manifolds and tabular benchmarks. Future work includes extending the framework to learned embeddings, developing tighter certification schemes via adaptive decompositions and approximate Voronoi diagrams for scaling to higher dimensional data, and leveraging the geometric awareness for continual learning, controlled generation and interpretable anomaly detection using PCs.

(shaded region). As training proceeds and the temperature annealing sharpens the gates, the lower bound increases steadily and the envelope stabilizes reflecting improved fit of local experts and tighter normalization bounds as the learned partitions align with the data support. HFV remains tractable throughout, and its curve corresponds to exact likelihood under the aligned factorized gating. Overall, these results suggest that VT offers substantial gains while remaining certifiable, and that soft-gate training with temperature annealing yields stable learning dynamics.

### 4.2. Higher Dimensional Tabular Datasets

To assess the scalability of our constructions beyond the synthetic settings, we also consider four standard UCI tabular density estimation benchmarks: **power** ($D=6$), **gas** ($D=8$), **hepmass** ($D=21$), and **miniboone** ($D=43$) (Papamakarios et al., 2017; Loconte et al., 2024). Using the same base architecture of 40 input and sum units and an identical training protocol, we evaluate both EinsumNet and HCLT families using VT and HFV variants. VT models report certified lower and upper bounds without any additional refinement while HFV models use exact inference.

## Acknowledgements

The authors gratefully acknowledge the generous support by the AFOSR award FA9550-23-1-0239, the ARO award W911NF2010224 and the DARPA Assured Neuro Symbolic Learning and Reasoning (ANSR) award HR001122S0039.

## Impact Statement

This work develops theoretical foundations for geometry-aware probabilistic circuits, a class of tractable generative models that support exact and efficient probabilistic inference. The primary contributions are mathematical: formalizing an incompatibility between geometric routing and tractable inference, and identifying conditions under which it can be resolved. As such, the direct societal impact is limited at this stage. But on the positive side, more expressive tractable models can benefit applications that require reliable uncertainty quantification, such as medical diagnosis, scientific modeling, and safety-critical decision making, where the guarantees of exact inference are essential. The explicit geometric structure introduced by our approach also improves interpretability, as routing decisions correspond to identifiable regions of the input space, and can lead to auditability in high-stakes settings.

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

# A. Proofs of Main Results

## A.1. Intractability of Voronoi-Gated PCs

We begin by establishing why unconstrained Voronoi gating breaks tractable inference, even in the simplest case of a single sum node with fully factorized experts.

**Proposition A.1** (Single-Layer Intractability). *Let $f(\mathbf{x}) = \sum_{k=1}^{K} g_k(\mathbf{x})\, \pi_k\, p_k(\mathbf{x})$ be a Voronoi-gated sum node over variables $\mathbf{X} = \{X_1, \ldots, X_D\}$ where $g_k(\mathbf{x}) = \mathbb{I}[\mathbf{x} \in V_k]$ with Voronoi cells $\{V_k\}$ induced by centroids $\{\mathbf{c}_1, \ldots, \mathbf{c}_K\} \subset \mathbb{R}^D$. Suppose each expert is fully factorized: $p_k(\mathbf{x}) = \prod_{i=1}^{D} p_k^{(i)}(x_i)$. Then the partition function*

$$Z = \int f(\mathbf{x})\, d\mathbf{x} = \sum_{k=1}^{K} \pi_k \int_{V_k} p_k(\mathbf{x})\, d\mathbf{x}$$

*requires integrating $p_k$ over Voronoi cells $V_k$, which are convex polytopes with oblique boundaries. In general, the integral $\int_{V_k} \prod_i p_k^{(i)}(x_i)\, d\mathbf{x}$ does not factor into a product of one-dimensional integrals.*

*Proof.* Each Voronoi cell $V_k$ is defined as the region of points closer to centroid $\mathbf{c}_k$ than to any other centroid:

$$V_k = \left\{ \mathbf{x} \in \mathbb{R}^D : \|\mathbf{x} - \mathbf{c}_k\|^2 \leq \|\mathbf{x} - \mathbf{c}_j\|^2 \text{ for all } j \right\}.$$

Equivalently, $V_k$ is the intersection of $K - 1$ half-spaces:

$$V_k = \bigcap_{j \neq k} H_{kj}^{-},$$

where $H_{kj}^{-} = \{\mathbf{x} : (\mathbf{c}_j - \mathbf{c}_k)^{\top} \mathbf{x} \leq \frac{1}{2}(\|\mathbf{c}_j\|^2 - \|\mathbf{c}_k\|^2)\}$.

The boundary hyperplane between cells $V_k$ and $V_j$ is

$$H_{kj} = \left\{ \mathbf{x} : (\mathbf{c}_j - \mathbf{c}_k)^{\top} \mathbf{x} = \tfrac{1}{2}(\|\mathbf{c}_j\|^2 - \|\mathbf{c}_k\|^2) \right\}.$$

For generic centroids (i.e., centroids not satisfying special symmetries), the normal vector $\mathbf{c}_j - \mathbf{c}_k$ has multiple nonzero components. This means the boundary hyperplane is not aligned with any coordinate axis, it is *oblique* with respect to the standard coordinate system. Consider the integral:

$$\int_{V_k} \prod_{i=1}^{D} p_k^{(i)}(x_i)\, d\mathbf{x}.$$

Even though the integrand factors across variables, the region $V_k$ does not. Specifically, $V_k$ is not a Cartesian product of the form $I_1 \times \cdots \times I_D$ where each $I_i \subseteq \mathbb{R}$ is an interval. The oblique boundaries couple the coordinates: whether a point $\mathbf{x}$ lies in $V_k$ depends on the joint configuration of all coordinates, not on independent conditions on each coordinate separately.

To apply Fubini's theorem and factor the integral as $\prod_i \int_{I_i} p_k^{(i)}(x_i)\, dx_i$, we would need the integration domain to be a product of univariate domains. Since $V_k$ is not such a product, Fubini does not reduce the $D$-dimensional integral to a product of $D$ one-dimensional integrals.

**Concrete Example (2D)**: Let $D = 2$ with centroids $\mathbf{c}_1 = (0, 1)$ and $\mathbf{c}_2 = (1, 0)$. The Voronoi boundary between $V_1$ and $V_2$ satisfies

$$\|(x_1, x_2) - (0, 1)\|^2 = \|(x_1, x_2) - (1, 0)\|^2,$$

which simplifies to $x_1^2 + (x_2 - 1)^2 = (x_1 - 1)^2 + x_2^2$, yielding $x_1 = x_2$.

Thus $V_1 = \{(x_1, x_2) : x_2 > x_1\}$ (the region above the diagonal). Now consider

$$\int_{V_1} p_1^{(1)}(x_1) p_1^{(2)}(x_2)\, dx_1\, dx_2.$$

Integrating first over $x_2$ for fixed $x_1$:

$$\int_{V_1} p_1^{(1)}(x_1) p_1^{(2)}(x_2)\, dx_1\, dx_2 = \int_{-\infty}^{\infty} p_1^{(1)}(x_1) \left( \int_{x_1}^{\infty} p_1^{(2)}(x_2)\, dx_2 \right) dx_1.$$

The inner integral has lower limit $x_1$, which depends on the outer integration variable. This prevents factorization into $\left( \int p_1^{(1)}(x_1) dx_1 \right) \left( \int p_1^{(2)}(x_2) dx_2 \right)$, and the dependence persists regardless of how simple the univariate densities $p_1^{(i)}$ are. Thus, the oblique geometry of Voronoi cells couples variables in a way that is fundamentally incompatible with the factorization exploited by decomposable probabilistic circuits. $\square$

### A.1.1. DEEP CIRCUITS AND CONSTRAINT ACCUMULATION

**Proposition A.2** (Constraint Accumulation). *In a deep circuit of depth $L$ with Voronoi gating at multiple levels, each contribution is integrated over a region determined by the active Voronoi cell at every ancestor sum node. The feasible set is an intersection of Voronoi regions restricted to different scopes.*

At the root, routing selects $V_{k_{\text{root}}}$ over full scope $\mathbf{X}$. At an internal node with scope $S \subset \mathbf{X}$, routing selects $V_{k_S}$ over $\mathbf{X}_S$. After marginalizing out variables not in $S$, the root constraint projects onto $\mathbf{X}_S$ in a way that generally doesn't align with $V_{k_S}$. The resulting region is an intersection of projected polytopes, which can be arbitrarily complex.

**Example A.3** (Two-Level Interaction). *Consider vtree root scope $\{X_1, X_2, X_3, X_4\}$ with children $\{X_1, X_2\}$ and $\{X_3, X_4\}$.*

- *Root gate imposes $x_1 + x_2 + x_3 + x_4 \le c_1$.*

- *Left child gate imposes $x_1 - x_2 \le c_2$.*

*To integrate over $\{X_1, X_2\}$, we need both the local constraint $x_1 - x_2 \le c_2$ and the projected root constraint. After marginalizing out $\{X_3, X_4\}$, the root constraint becomes context-dependent, preventing independent marginalization.*

Thus, in deep circuits, unconstrained Voronoi gating introduces hierarchical geometric constraints that interact in complex ways after marginalization, compounding intractability. HFV-PCs avoid this by ensuring geometric constraints factor along circuit variable partitions at every level.

### A.2. Geometric Alignment and Tractability Recovery

The preceding result shows that unconstrained Voronoi cells break tractability. We now prove that alignment with the circuit's variable decomposition is sufficient to restore it.

**Theorem A.4** (Alignment Enables Factorization). *Consider a gated sum node $f(\mathbf{x}_S) = \sum_{k=1}^{K} g_k(\mathbf{x}_S)\, \pi_k\, p_k(\mathbf{x}_S)$ over scope $\mathbf{X}_S = \mathbf{X}_{S_1} \sqcup \mathbf{X}_{S_2}$ (disjoint partition). Suppose each gating region $R_k$ satisfies the alignment condition: there exist $R_k^{(1)} \subseteq \mathbb{R}^{|\mathbf{X}_{S_1}|}$ and $R_k^{(2)} \subseteq \mathbb{R}^{|\mathbf{X}_{S_2}|}$ such that*

$$R_k = R_k^{(1)} \times R_k^{(2)},$$

*and suppose each expert factors as $p_k(\mathbf{x}_S) = p_k^{(1)}(\mathbf{x}_{S_1}) p_k^{(2)}(\mathbf{x}_{S_2})$. Then the partition function decomposes:*

$$\int f(\mathbf{x}_S)\, d\mathbf{x}_S = \sum_{k=1}^{K} \pi_k \left( \int_{R_k^{(1)}} p_k^{(1)}(\mathbf{x}_{S_1})\, d\mathbf{x}_{S_1} \right) \left( \int_{R_k^{(2)}} p_k^{(2)}(\mathbf{x}_{S_2})\, d\mathbf{x}_{S_2} \right).$$

*Proof.* Start with the definition of $f$ and substitute the gate and expert factorizations:

$$\int f(\mathbf{x}_S)\, d\mathbf{x}_S = \int \sum_{k=1}^{K} \mathbb{I}[\mathbf{x}_S \in R_k]\, \pi_k\, p_k(\mathbf{x}_S)\, d\mathbf{x}_S$$

$$= \sum_{k=1}^{K} \pi_k \int_{R_k} p_k(\mathbf{x}_S)\, d\mathbf{x}_S.$$

By the alignment condition, $\mathbf{x}_S \in R_k$ if and only if $\mathbf{x}_{S_1} \in R_k^{(1)}$ and $\mathbf{x}_{S_2} \in R_k^{(2)}$ simultaneously. Since $\mathbf{X}_{S_1}$ and $\mathbf{X}_{S_2}$ are disjoint, we can write $d\mathbf{x}_S = d\mathbf{x}_{S_1} d\mathbf{x}_{S_2}$ and apply Fubini's theorem:

$$\int_{R_k} p_k(\mathbf{x}_S) \, d\mathbf{x}_S = \int_{R_k^{(1)} \times R_k^{(2)}} p_k^{(1)}(\mathbf{x}_{S_1}) p_k^{(2)}(\mathbf{x}_{S_2}) \, d\mathbf{x}_{S_1} \, d\mathbf{x}_{S_2}$$

$$= \left( \int_{R_k^{(1)}} p_k^{(1)}(\mathbf{x}_{S_1}) \, d\mathbf{x}_{S_1} \right) \left( \int_{R_k^{(2)}} p_k^{(2)}(\mathbf{x}_{S_2}) \, d\mathbf{x}_{S_2} \right).$$

Substituting back completes the proof. $\square$

**Remark A.5.** *This result generalizes immediately to $m$-way partitions $\mathbf{X}_S = \bigsqcup_{i=1}^m \mathbf{X}_{S_i}$ with regions $R_k = \prod_{i=1}^m R_k^{(i)}$ and experts $p_k(\mathbf{x}_S) = \prod_i p_k^{(i)}(\mathbf{x}_{S_i})$. The key requirement is that both the gating regions and the expert distributions respect the same factorization structure.*

### A.3. Certified Approximate Inference

A.3.1. INNER BOX CONSTRUCTION

**Proposition A.6** (Conservative Inner Box). *Let $\{V_k\}$ be Voronoi cells induced by centroids $\{\mathbf{c}_k\} \subset \mathbb{R}^d$. Define the nearest-centroid distance*

$$\delta_k := \min_{j \neq k} \|\mathbf{c}_k - \mathbf{c}_j\|_2.$$

*Then the axis-aligned box centered at $\mathbf{c}_k$ with radius*

$$r = \frac{\delta_k}{2\sqrt{d}}$$

*satisfies $B_k^- := \prod_{i=1}^d [\mathbf{c}_{k,i} - r, \mathbf{c}_{k,i} + r] \subseteq V_k$.*

*Proof.* Take any point $\mathbf{x} \in B_k^-$. By construction, $|x_i - \mathbf{c}_{k,i}| \leq r$ for all $i$, so

$$\|\mathbf{x} - \mathbf{c}_k\|_2 = \sqrt{\sum_{i=1}^d (x_i - \mathbf{c}_{k,i})^2} \leq \sqrt{d \cdot r^2} = \sqrt{d} \, r = \frac{\delta_k}{2}.$$

Now consider any other centroid $\mathbf{c}_j$ with $j \neq k$. By the triangle inequality:

$$\|\mathbf{x} - \mathbf{c}_j\|_2 \geq \|\mathbf{c}_k - \mathbf{c}_j\|_2 - \|\mathbf{x} - \mathbf{c}_k\|_2 \geq \delta_k - \frac{\delta_k}{2} = \frac{\delta_k}{2}.$$

Thus $\|\mathbf{x} - \mathbf{c}_k\|_2 \leq \frac{\delta_k}{2} \leq \|\mathbf{x} - \mathbf{c}_j\|_2$ for all $j \neq k$, which means $\mathbf{x}$ is closer to $\mathbf{c}_k$ than to any other centroid. By the definition of Voronoi cells, $\mathbf{x} \in V_k$. Since this holds for every $\mathbf{x} \in B_k^-$, we have $B_k^- \subseteq V_k$. $\square$

**Remark A.7.** *The factor $1/(2\sqrt{d})$ ensures containment but is conservative. A tighter construction can be obtained by optimizing the per-dimension radii $r_i$ subject to the half-space constraints defining $V_k$, but this requires solving a constrained optimization problem for each cell. The simple formula above trades tightness for computational convenience and robustness.*

**Remark A.8** (Extension to bounded domains). *If the domain is a box $\Omega = \prod_i [\ell_i, u_i]$, intersect: $B_k^- \leftarrow B_k^- \cap \Omega$. This preserves $B_k^- \subseteq V_k \cap \Omega$ because intersection with $\Omega$ only removes points outside the domain.*

A.3.2. ANYTIME REFINEMENT: CONVERGENCE ANALYSIS

**Theorem A.9** (Monotone Tightening and Convergence). *Let $\mathcal{P}_t$ denote the partition after $t$ refinement steps with bounds $(Z_t^-, Z_t^+)$. Then (i) $Z_t^- \leq Z \leq Z_t^+ \, \forall \, t$ (ii) $Z_t^- \leq Z_{t+1}^-$ and $Z_{t+1}^+ \leq Z_t^+$ for all $t$ (iii) if refinement drives the boundary volume $\mu(V_k^+(\mathcal{P}_t) \setminus V_k^-(\mathcal{P}_t)) \to 0$ for each cell, then $\lim_{t \to \infty} Z_t^{\pm} = Z$. Under uniform refinement, the gap scales as $Z_t^+ - Z_t^- = O(2^{-t/d})(Z_0^+ - Z_0^-)$, requiring depth $O(d \log(1/\epsilon))$ to achieve target gap $\epsilon$.*

*Proof.* We formalize the refinement scheme and then prove each claim. Fix a Voronoi-gated sum node $n$ with scope $S$ and bounded domain $\Omega_S \subset \mathbb{R}^{|S|}$. Let $\{V_k\}_{k=1}^K$ be the Voronoi cells intersected with $\Omega_S$ (so each $V_k$ is measurable and $\bigcup_k V_k = \Omega_S$ up to boundaries of measure zero). At refinement step $t$, we maintain a *disjoint* axis-aligned partition $\mathcal{P}_t$ of $\Omega_S$ into boxes, i.e., $\Omega_S = \biguplus_{B \in \mathcal{P}_t} B$ (disjoint union). For each cell $k$, define the inner/outer approximations induced by $\mathcal{P}_t$:

$$V_k^-(\mathcal{P}_t) := \bigcup_{B \in \mathcal{P}_t:\, B \subseteq V_k} B, \qquad\qquad V_k^+(\mathcal{P}_t) := \bigcup_{B \in \mathcal{P}_t:\, B \cap V_k \neq \emptyset} B. \tag{3}$$

By construction, $V_k^-(\mathcal{P}_t) \subseteq V_k \subseteq V_k^+(\mathcal{P}_t)$ for every $t$.

Let $p_k(\mathbf{x}_S) \geq 0$ denote the expert density (subcircuit) attached to cell $k$. Define the node-level cell integrals and their bounds

$$I_k := \int_{V_k} p_k(\mathbf{x}_S)\, d\mathbf{x}_S, \qquad I_{k,t}^- := \int_{V_k^-(\mathcal{P}_t)} p_k(\mathbf{x}_S)\, d\mathbf{x}_S, \qquad I_{k,t}^+ := \int_{V_k^+(\mathcal{P}_t)} p_k(\mathbf{x}_S)\, d\mathbf{x}_S. \tag{4}$$

Because $\mathcal{P}_t$ is disjoint and $V_k^-(\mathcal{P}_t)$ (resp. $V_k^+(\mathcal{P}_t)$) is a union of boxes from $\mathcal{P}_t$, we can write these integrals as sums of box integrals (no overlap):

$$I_{k,t}^- = \sum_{B \in \mathcal{P}_t:\, B \subseteq V_k} \text{INTEGRATEBOX}(p_k, B),$$

$$I_{k,t}^+ = \sum_{B \in \mathcal{P}_t:\, B \cap V_k \neq \emptyset} \text{INTEGRATEBOX}(p_k, B).$$

The node-level mixture integral at $n$ is $I := \sum_{k=1}^K \pi_k I_k$ and its bounds are $I_t^- := \sum_k \pi_k I_{k,t}^-$ and $I_t^+ := \sum_k \pi_k I_{k,t}^+$, where $\pi_k \geq 0$ and $\sum_k \pi_k = 1$. Circuit-level bounds $(Z_t^-, Z_t^+)$ are obtained by propagating node-level bounds upward via the same sum/product rules as in Theorem 3.10.

**Claim (i): Validity.** $Z_t^- \leq Z \leq Z_t^+$. We first establish node-level validity at every Voronoi-gated node $n$. For each $k$, since $V_k^-(\mathcal{P}_t) \subseteq V_k \subseteq V_k^+(\mathcal{P}_t)$ and $p_k \geq 0$, monotonicity of integration gives

$$I_{k,t}^- \leq I_k \leq I_{k,t}^+.$$

Multiplying by $\pi_k \geq 0$ and summing over $k$ yields $I_t^- \leq I \leq I_t^+$ at node $n$.

For standard sum nodes, linearity and nonnegative weights preserve inequalities; for product nodes, decomposability implies integrals factor and multiplying nonnegative bounds preserves inequalities. Thus, by induction in reverse topological order (as in Theorem 3.10), every node integral is sandwiched by its bounds, in particular at the root $Z_t^- \leq Z \leq Z_t^+$.

**Claim (ii): Monotonic tightening.** By definition, $\mathcal{P}_{t+1}$ is obtained from $\mathcal{P}_t$ by refining (bisecting) a subset of boxes and reclassifying. Refinement preserves disjointness and refines the partition: every box $B' \in \mathcal{P}_{t+1}$ is contained in some box $B \in \mathcal{P}_t$, and every $B \in \mathcal{P}_t$ is the disjoint union of its descendants in $\mathcal{P}_{t+1}$. We show set inclusions:

$$V_k^-(\mathcal{P}_t) \subseteq V_k^-(\mathcal{P}_{t+1}), \qquad V_k^+(\mathcal{P}_{t+1}) \subseteq V_k^+(\mathcal{P}_t). \tag{5}$$

For the inner sets: take any $B \in \mathcal{P}_t$ with $B \subseteq V_k$. Under refinement, $B$ is either unchanged or replaced by disjoint children $\{B_r'\} \subseteq B$. In either case, every descendant $B_r'$ satisfies $B_r' \subseteq B \subseteq V_k$, hence all of $B$'s mass remains INSIDE and is included in $V_k^-(\mathcal{P}_{t+1})$. Therefore the union of INSIDE boxes can only expand, proving the first inclusion in (5).

For the outer sets: take any $B' \in \mathcal{P}_{t+1}$ such that $B' \cap V_k \neq \emptyset$. Let $B \in \mathcal{P}_t$ be its (unique) ancestor box with $B' \subseteq B$. Then necessarily $B \cap V_k \neq \emptyset$ as well, hence $B \subseteq V_k^+(\mathcal{P}_t)$. Since $B'$ is contained in $B$, we have $B' \subseteq V_k^+(\mathcal{P}_t)$, and taking unions over all such $B'$ gives the second inclusion in (5).

Now apply monotonicity of integration (using $p_k \geq 0$) to (5):

$$I_{k,t}^- = \int_{V_k^-(\mathcal{P}_t)} p_k \leq \int_{V_k^-(\mathcal{P}_{t+1})} p_k = I_{k,t+1}^-$$

$$I_{k,t+1}^+ = \int_{V_k^+(\mathcal{P}_{t+1})} p_k \le \int_{V_k^+(\mathcal{P}_t)} p_k = I_{k,t}^+.$$

Weighting by $\pi_k \ge 0$ and summing yields node-level monotonicity $I_t^- \le I_{t+1}^-$ and $I_{t+1}^+ \le I_t^+$. Finally, the circuit-level bounds follow by propagating inequalities through the circuit: standard sums preserve monotonicity by linearity with nonnegative weights, and products preserve monotonicity because all integrals are nonnegative and multiplication is monotone in each argument. Hence $Z_t^- \le Z_{t+1}^-$ and $Z_{t+1}^+ \le Z_t^+$.

**Claim (iii): Convergence under vanishing boundary volume.** Fix a Voronoi-gated node $n$ and a cell $k$. Define the *undecided* region

$$U_{k,t} := V_k^+(\mathcal{P}_t) \setminus V_k^-(\mathcal{P}_t),$$

so that $V_k^-(\mathcal{P}_t) \subseteq V_k \subseteq V_k^+(\mathcal{P}_t)$ implies

$$0 \le I_{k,t}^+ - I_{k,t}^- = \int_{U_{k,t}} p_k(\mathbf{x}_S)\, d\mathbf{x}_S. \tag{6}$$

Assume that the boundary volume $\mu(U_{k,t}) \to 0$ as $t \to \infty$. Since $p_k$ is integrable over $\Omega_S$ (it is a density component on a bounded domain), the Lebesgue integral is absolutely continuous with respect to the measure: for any $\varepsilon > 0$ there exists $\delta > 0$ such that $\mu(A) < \delta$ implies $\int_A p_k < \varepsilon$. Applying this to $A = U_{k,t}$ yields $\int_{U_{k,t}} p_k \to 0$, hence $I_{k,t}^+ - I_{k,t}^- \to 0$ by (6). Therefore $I_{k,t}^- \to I_k$ and $I_{k,t}^+ \to I_k$ (squeezing with validity from (i)).

Multiplying by $\pi_k$ and summing over $k$ shows that the node-level bounds converge to the true node integral. Applying the same inductive argument up the circuit—using continuity of addition and multiplication on $\mathbb{R}_{\ge 0}$ and the fact that all intermediate quantities are finite—yields convergence at the root: $Z_t^- \to Z$ and $Z_t^+ \to Z$.

The convergence-rate statement $Z_t^+ - Z_t^- = O(2^{-t/d})(Z_0^+ - Z_0^-)$ under uniform refinement is an *informal* geometric bound that depends on regularity of the density and the surface-area-to-volume behavior of Voronoi facets. A sufficient condition is, for example, that each expert density is bounded on $\Omega_S$ by $\|p_k\|_\infty < \infty$ and that the undecided region $U_{k,t}$ lies in a tubular neighborhood of the cell boundary whose thickness scales with the maximum box diameter. Under uniform refinement, after $t$ steps the maximum box side length is $O(L \cdot 2^{-t/d})$ where $L$ is the initial domain diameter. The undecided region $U_{k,t}$ lies in a neighborhood of thickness $O(2^{-t/d})$ around the $(d-1)$-dimensional boundary $\partial V_k$. Thus

$$\mu(U_{k,t}) = O(\text{surface area} \times \text{thickness}) = O(2^{-t/d}).$$

Assuming bounded density $\|p_k\|_\infty < \infty$:

$$I_{k,t}^+ - I_{k,t}^- \le \|p_k\|_\infty \cdot \mu(U_{k,t}) = O(2^{-t/d}).$$

Summing over $k$ and propagating through the circuit gives

$$Z_t^+ - Z_t^- = O(2^{-t/d})(Z_0^+ - Z_0^-).$$

To achieve gap $\epsilon$, we need $t \ge d \log_2((Z_0^+ - Z_0^-)/\epsilon) = O(d \log(1/\epsilon))$. $\qquad\square$

**Remark A.10** (Curse of dimensionality). *The factor $d$ in the refinement depth makes certified bounds impractical in high dimensions. For example, achieving $\epsilon = 0.01$ in $d = 10$ requires roughly $10 \log_2(100) \approx 66$ refinement levels, leading to exponential blowup. This motivates the HFV-PC construction in Section 3.2.*

A.3.3. COMPUTATIONAL COMPLEXITY OF REFINEMENT

**Proposition A.11** (Per-Iteration Complexity). *Each iteration of Algorithm 1 requires:*

- $O(|\mathcal{C}| \cdot K_{\max})$ *to compute certified bounds via bottom-up propagation.*

- $O(K \cdot d \cdot 2^d)$ *to test box-polytope containment/intersection for $K$ cells in dimension $d$.*

*Total per-iteration cost:* $O(|\mathcal{C}| \cdot K_{\max} + K \cdot d \cdot 2^d)$.

*Proof.* **Bound computation**: The circuit has $|\mathcal{C}|$ nodes. At each Voronoi-gated node, we integrate the expert subcircuit over at most $K$ boxes. Assuming box integration over a subcircuit is linear in subcircuit size, the total cost is $O(|\mathcal{C}| \cdot K_{\max})$.

**Box classification**: To determine whether box $B = \prod_i [l_i, u_i] \subseteq \mathbb{R}^d$ satisfies $B \subseteq V_k$ or $B \cap V_k = \emptyset$, we use the half-space representation:

$$V_k = \bigcap_{j \neq k} \{\mathbf{x} : (\mathbf{c}_j - \mathbf{c}_k)^\top \mathbf{x} \leq c_{jk}\}.$$

For each half-space, compute extremal values of $a^\top \mathbf{x}$ over box $B$. This takes $O(d)$ per half-space. There are $K - 1$ half-spaces, so testing one box against one cell takes $O(K \cdot d)$. Testing all $K$ cells takes $O(K^2 \cdot d)$. Alternatively, testing all $2^d$ corners of $B$ against all $K$ centroids takes $O(2^d \cdot K \cdot d)$. The dominant term is $O(2^d)$ for large $d$. □

**Remark A.12.** *The $2^d$ factor in corner evaluations reinforces the curse of dimensionality. In practice, for $d \leq 5$, adaptive refinement is effective; for higher dimensions, HFV-PCs are preferable.*

### A.3.4. MARGINALS AND CONDITIONALS

The certified bound framework extends naturally to marginals and conditionals.

**Marginal bounds.** To compute $p(\mathbf{x}_A) = \int p(\mathbf{x}) \, d\mathbf{x}_{\bar{A}}$ where $\bar{A} = \mathbf{X} \setminus \mathbf{X}_A$, propagate bounds through the circuit while marginalizing out $\bar{A}$. At each Voronoi-gated node:

- If the scope $S$ includes variables in $\bar{A}$, restrict box approximations to dimensions being integrated (project boxes onto subspace $\mathbf{X}_{\bar{A} \cap S}$) and integrate over those dimensions.

- If the scope is entirely within $A$, no change needed.

The result is bounds $(p^-(\mathbf{x}_A), p^+(\mathbf{x}_A))$ satisfying $p^-(\mathbf{x}_A) \leq p(\mathbf{x}_A) \leq p^+(\mathbf{x}_A)$ for all $\mathbf{x}_A$.

**Conditional bounds.** For disjoint $A, B$ and $p(\mathbf{x}_A \mid \mathbf{x}_B) = p(\mathbf{x}_A, \mathbf{x}_B)/p(\mathbf{x}_B)$, we have bounds on numerator and denominator. Assuming $p^-(\mathbf{x}_B) > 0$, apply interval arithmetic:

$$\frac{p^-(\mathbf{x}_A, \mathbf{x}_B)}{p^+(\mathbf{x}_B)} \leq p(\mathbf{x}_A \mid \mathbf{x}_B) \leq \frac{p^+(\mathbf{x}_A, \mathbf{x}_B)}{p^-(\mathbf{x}_B)}.$$

These bounds are valid because division is monotone in the numerator and antitone in the denominator when all quantities are positive.

**Practical considerations.** If $p^-(\mathbf{x}_B)$ is very small or zero, the upper bound becomes loose or undefined. Refining the partition to tighten $p^-(\mathbf{x}_B)$ away from zero is necessary for meaningful conditional bounds.

### A.4. Hierarchical Factorized Voronoi PCs

#### A.4.1. FACTORIZED VORONOI CELLS

The key idea of HFV-PCs is to replace a single high-dimensional Voronoi tessellation with a *product* of lower-dimensional tessellations that align with the circuit's variable decomposition.

**Definition A.13** (Factorized Voronoi Partition). *Let $\mathbf{X}_S = \bigsqcup_{i=1}^m \mathbf{X}_{S_i}$ be a partition of scope $S$ into $m$ disjoint blocks. For each block $i$, choose centroids $\{\mathbf{c}_1^{(i)}, \ldots, \mathbf{c}_{K_i}^{(i)}\} \subset \mathbb{R}^{|\mathbf{X}_{S_i}|}$ and let $\{V_{k_i}^{(i)}\}_{k_i=1}^{K_i}$ be the induced Voronoi cells in $\mathbb{R}^{|\mathbf{X}_{S_i}|}$. Define the* joint product cell *indexed by $\mathbf{k} = (k_1, \ldots, k_m) \in [K_1] \times \cdots \times [K_m]$:*

$$V_{\mathbf{k}} := V_{k_1}^{(1)} \times \cdots \times V_{k_m}^{(m)} \subseteq \mathbb{R}^{|\mathbf{X}_S|}.$$

*The corresponding* hard gate *factors as:*

$$g_{\mathbf{k}}(\mathbf{x}_S) = \mathbb{I}[\mathbf{x}_S \in V_{\mathbf{k}}] = \prod_{i=1}^{m} \mathbb{I}[\mathbf{x}_{S_i} \in V_{k_i}^{(i)}] = \prod_{i=1}^{m} g_{k_i}^{(i)}(\mathbf{x}_{S_i}).$$

Membership in $V_{\mathbf{k}}$ can be decided independently for each block. This is exactly the geometric analogue of decomposability.

**Proposition A.14** (Factorized Gate Decomposition). *For a factorized cell $V_{\mathbf{k}}$, the hard gate decomposes as*

$$g_{\mathbf{k}}(\mathbf{x}_S) = \prod_{i=1}^{m} g_{k_i}^{(i)}(\mathbf{x}_{S_i}).$$

*Proof.* By definition, $\mathbf{x}_S \in V_{\mathbf{k}}$ if and only if $\mathbf{x}_{S_i} \in V_{k_i}^{(i)}$ for all $i$. Since the conditions involve disjoint variable sets, the indicator of their conjunction factors:

$$\mathbb{I}[\mathbf{x}_S \in V_{\mathbf{k}}] = \mathbb{I}\left[\bigcap_i \{\mathbf{x}_{S_i} \in V_{k_i}^{(i)}\}\right] = \prod_i \mathbb{I}[\mathbf{x}_{S_i} \in V_{k_i}^{(i)}].$$

$\square$

### A.4.2. HFV-GATED SUM NODES AND TRACTABILITY

**Theorem A.15** (Tractability of HFV-PCs). *Let $\mathcal{C}$ be an HFV-PC with $|\mathcal{C}|$ nodes, maximum factorization degree $m$, and at most $K$ Voronoi cells per factor. Then the partition function, all marginals, and all conditionals are computable exactly in time $O(|\mathcal{C}|K^m)$.*

*Proof.* We prove the partition function claim. The marginal and conditional claims follow by the same bottom-up evaluation used for standard smooth decomposable PCs.

For each node $n$ we consider $I_n = \int f_n(\mathbf{x}_{\text{scope}(n)}) \, d\mathbf{x}_{\text{scope}(n)}$.

If $n$ is a leaf, $I_n$ is a univariate integral and is tractable by assumption.

If $n$ is a product node, decomposability implies that children have disjoint scopes, so the integral factors as $I_n = \prod_{c \in \text{ch}(n)} I_c$.

If $n$ is a standard sum node, linearity gives $I_n = \sum_c \pi_c I_c$.

It remains to handle an HFV-gated sum node $n$ with scope $\mathbf{X}_S = \bigsqcup_{i=1}^{m} \mathbf{X}_{S_i}$. Using (**??**) and exchanging summation and integration we obtain

$$I_n = \int \sum_{\mathbf{k}} g_{\mathbf{k}}(\mathbf{x}_S) \, \pi_{\mathbf{k}} \prod_{i=1}^{m} p_{k_i}^{(i)}(\mathbf{x}_{S_i}) \, d\mathbf{x}_S$$

$$= \sum_{\mathbf{k}} \pi_{\mathbf{k}} \int \prod_{i=1}^{m} g_{k_i}^{(i)}(\mathbf{x}_{S_i}) \, p_{k_i}^{(i)}(\mathbf{x}_{S_i}) \, d\mathbf{x}_S. \tag{7}$$

Since the scopes $\mathbf{X}_{S_i}$ are disjoint and each factor depends only on $\mathbf{x}_{S_i}$, we apply Fubini to factor the integral,

$$\int \prod_{i=1}^{m} g_{k_i}^{(i)}(\mathbf{x}_{S_i}) \, p_{k_i}^{(i)}(\mathbf{x}_{S_i}) \, d\mathbf{x}_S \tag{8}$$

$$= \prod_{i=1}^{m} \int g_{k_i}^{(i)}(\mathbf{x}_{S_i}) \, p_{k_i}^{(i)}(\mathbf{x}_{S_i}) \, d\mathbf{x}_{S_i}. \tag{9}$$

Substituting $g_{k_i}^{(i)}(\mathbf{x}_{S_i}) = \mathbb{I}[\mathbf{x}_{S_i} \in V_{k_i}^{(i)}]$ yields

$$I_n = \sum_{\mathbf{k}} \pi_{\mathbf{k}} \prod_{i=1}^{m} I_{k_i}^{(i)}, \qquad I_{k_i}^{(i)} = \int_{V_{k_i}^{(i)}} p_{k_i}^{(i)}(\mathbf{x}_{S_i}) \, d\mathbf{x}_{S_i}.$$

Each $I_{k_i}^{(i)}$ is the integral of an HFV subcircuit over a lower-dimensional Voronoi cell, so we compute it recursively by the same argument. This establishes exact tractability.

For complexity, at an HFV-gated sum node we compute $\sum_i K_i$ factor integrals and then evaluate the sum over at most $\prod_i K_i \le K^m$ joint indices, each term requiring $O(m)$ arithmetic operations. Summing over all circuit nodes yields $O(|\mathcal{C}|K^m)$ time. $\qquad\square$

**Cell-Restricted Integrals and the Recursion Base Case**   The recursion requires that we can evaluate integrals of the form $\int_{V_{k_i}^{(i)}} p_{k_i}^{(i)}$ at every level. In an HFV-PC the same factorization property ensures that cell restrictions are handled locally at the appropriate scope. The recursion bottoms out at univariate leaves, where Voronoi cells are intervals and integration is straightforward.

**Proposition A.16** (Univariate Voronoi Cells). *In $\mathbb{R}^1$, let $c_1 < \cdots < c_K$ be centroids. The induced Voronoi tessellation consists of intervals*

$$V_k = \begin{cases} (-\infty, \frac{c_1+c_2}{2}] & k = 1 \\ (\frac{c_{k-1}+c_k}{2}, \frac{c_k+c_{k+1}}{2}] & 1 < k < K \\ (\frac{c_{K-1}+c_K}{2}, \infty) & k = K. \end{cases}$$

For standard univariate leaf families such as Gaussians, mixtures of Gaussians, exponentials, and bounded distributions, integrating over an interval is tractable by closed form CDF evaluation or simple numerical routines.

**Definition A.17** (Binary HFV-PC). *A binary HFV-PC over variables $\mathbf{X}$ with vtree $\mathcal{T}$ is a probabilistic circuit in which each internal vtree node $v$ with children $v_L$ and $v_R$ induces HFV-gated sum nodes over scope $\mathbf{X}_v = \mathbf{X}_{v_L} \sqcup \mathbf{X}_{v_R}$. Each HFV-gated sum uses the binary partition $(\mathbf{X}_{v_L}, \mathbf{X}_{v_R})$ and computes:*

$$f(\mathbf{x}_v) = \sum_{k_L, k_R} g_{k_L}^{(L)}(\mathbf{x}_{v_L})\, g_{k_R}^{(R)}(\mathbf{x}_{v_R})\, \pi_{k_L, k_R}\, p_{k_L}^{(L)}(\mathbf{x}_{v_L})\, p_{k_R}^{(R)}(\mathbf{x}_{v_R}).$$

## A.5. Learning via Soft Gating

### A.5.1. DIFFERENTIABILITY AND GRADIENTS

**Proposition A.18** (Differentiability). *Let $\mathcal{C}$ be a soft Voronoi gated PC with parameters $\Theta = \{\{\mathbf{c}_{k_i}^{(i)}\}, \{\pi_\mathbf{k}\}, \theta_{leaf}\}$. For any finite temperature $\alpha > 0$, the likelihood $p(\mathbf{x}; \Theta, \alpha)$ is differentiable with respect to all parameters.*

*Proof.* Each soft gate

$$w_k(\mathbf{u}; \alpha) = \frac{\exp(-\alpha\|\mathbf{u} - \mathbf{c}_k\|^2)}{\sum_j \exp(-\alpha\|\mathbf{u} - \mathbf{c}_j\|^2)}$$

is a composition of smooth operations: squared Euclidean distance (polynomial), exponential, and softmax normalization (ratio of positive smooth functions). The circuit output is built from sums and products of smooth gates and leaf densities, hence differentiable in all parameters. $\qquad\square$

**Proposition A.19** (Centroid Gradient). *For soft gate $w_k(\mathbf{u}; \alpha)$, the gradient with respect to centroid $\mathbf{c}_k$ is:*

$$\nabla_{\mathbf{c}_k} w_k(\mathbf{u}; \alpha) = 2\alpha\, w_k(\mathbf{u}; \alpha)\big(1 - w_k(\mathbf{u}; \alpha)\big)(\mathbf{u} - \mathbf{c}_k).$$

*Proof.* Let $d_j = \|\mathbf{u} - \mathbf{c}_j\|^2$ and $Z = \sum_j \exp(-\alpha d_j)$. Then $w_k = \exp(-\alpha d_k)/Z$.

Taking the derivative with respect to $\mathbf{c}_k$:

$$\nabla_{\mathbf{c}_k} w_k = \frac{\nabla_{\mathbf{c}_k}[\exp(-\alpha d_k)]}{Z} - \frac{\exp(-\alpha d_k)\, \nabla_{\mathbf{c}_k}[Z]}{Z^2}.$$

We have $\nabla_{\mathbf{c}_k} d_k = -2(\mathbf{u} - \mathbf{c}_k)$, so:

$$\nabla_{\mathbf{c}_k}[\exp(-\alpha d_k)] = 2\alpha \exp(-\alpha d_k)(\mathbf{u} - \mathbf{c}_k).$$

Since only the $k$-th term in $Z$ depends on $\mathbf{c}_k$:

$$\nabla_{\mathbf{c}_k}[Z] = 2\alpha \exp(-\alpha d_k)(\mathbf{u} - \mathbf{c}_k).$$

Substituting:

$$\nabla_{\mathbf{c}_k} w_k = \frac{2\alpha \exp(-\alpha d_k)(\mathbf{u} - \mathbf{c}_k)}{Z} - \frac{\exp(-\alpha d_k) \cdot 2\alpha \exp(-\alpha d_k)(\mathbf{u} - \mathbf{c}_k)}{Z^2}$$
$$= 2\alpha\, w_k\, (1 - w_k)\, (\mathbf{u} - \mathbf{c}_k).$$

$\square$

The factor $w_k(1 - w_k)$ is largest when $w_k \approx 1/2$ (near decision boundaries where the model is uncertain). The gradient magnitude peaks in regions of ambiguity and vanishes where routing is confident, naturally focusing centroid updates on improving contested region geometry.

### A.5.2. SOFT-TO-HARD CONVERGENCE

**Theorem A.20** (Soft-to-Hard Convergence). *Let $\{V_k\}$ be Voronoi cells induced by centroids $\{\mathbf{c}_k\}$ and let $g_k(\mathbf{u}) = \mathbb{I}[\mathbf{u} \in V_k]$ be the hard gate. Let $w_k(\mathbf{u}; \alpha)$ be the soft gate. Then:*

*(i) For any $\mathbf{u}$ not on a Voronoi boundary, $\lim_{\alpha \to \infty} w_k(\mathbf{u}; \alpha) = g_k(\mathbf{u})$.*

*(ii) If $k^*(\mathbf{u}) := \arg\min_j \|\mathbf{u} - \mathbf{c}_j\|$ and margin $\gamma(\mathbf{u}) := \min_{j \neq k^*}(\|\mathbf{u} - \mathbf{c}_j\|^2 - \|\mathbf{u} - \mathbf{c}_{k^*}\|^2) > 0$, then*

$$1 - w_{k^*}(\mathbf{u}; \alpha) \leq (K - 1) \exp(-\alpha\gamma(\mathbf{u})).$$

*(iii) If $p$ is integrable, then $\lim_{\alpha \to \infty} \int w_k(\mathbf{u}; \alpha) p(\mathbf{u})\, d\mathbf{u} = \int_{V_k} p(\mathbf{u})\, d\mathbf{u}$.*

*Proof.* **(i) Pointwise convergence**: Let $\mathbf{u}$ be a point not on any Voronoi boundary. Then there exists unique nearest centroid $\mathbf{c}_{k^*}$ with $d_{k^*} := \|\mathbf{u} - \mathbf{c}_{k^*}\|^2 < d_j := \|\mathbf{u} - \mathbf{c}_j\|^2$ for all $j \neq k^*$.

Rewrite:

$$w_{k^*}(\mathbf{u}; \alpha) = \frac{1}{1 + \sum_{j \neq k^*} \exp(-\alpha(d_j - d_{k^*}))}.$$

Since $d_j - d_{k^*} > 0$ for all $j \neq k^*$, as $\alpha \to \infty$, each term $\exp(-\alpha(d_j - d_{k^*})) \to 0$ exponentially. Therefore:

$$\lim_{\alpha \to \infty} w_{k^*}(\mathbf{u}; \alpha) = 1 = g_{k^*}(\mathbf{u}).$$

For $j \neq k^*$, we have $\lim_{\alpha \to \infty} w_j(\mathbf{u}; \alpha) = 0 = g_j(\mathbf{u})$.

**(ii) Exponential rate**: Define margin $\gamma(\mathbf{u}) = \min_{j \neq k^*}(d_j - d_{k^*}) > 0$. Then for all $j \neq k^*$:

$$\exp(-\alpha(d_j - d_{k^*})) \leq \exp(-\alpha\gamma(\mathbf{u})).$$

Summing:

$$\sum_{j \neq k^*} \exp(-\alpha(d_j - d_{k^*})) \leq (K - 1) \exp(-\alpha\gamma(\mathbf{u})).$$

Therefore:

$$w_{k^*} \geq \frac{1}{1 + (K - 1)\exp(-\alpha\gamma)},$$

implying

$$1 - w_{k^*} \leq (K - 1)\exp(-\alpha\gamma(\mathbf{u})).$$

---

**Algorithm 3** Soft Gating Training with Annealing

---

**Require:** Dataset $\mathcal{D} = \{\mathbf{x}^{(n)}\}_{n=1}^{N}$, schedule $\{\alpha_t\}_{t=1}^{T}$, learning rate $\eta$
**Ensure:** Trained parameters $\Theta$
 1: **Initialize:**
 2:  Centroids $\{\mathbf{c}_{k_i}^{(i)}\}$ via $k$-means (per factor for HFV)
 3:  Mixture weights $\{\pi_\mathbf{k}\}$ uniformly on simplex
 4:  Leaf parameters $\theta_{\text{leaf}}$ randomly or from prior
 5: **for** $t = 1$ to $T$ **do**
 6:  Set inverse temperature $\alpha \leftarrow \alpha_t$
 7:  **for** each minibatch $\mathcal{B} \subset \mathcal{D}$ **do**
 8:   $\mathcal{L} \leftarrow -\frac{1}{|\mathcal{B}|} \sum_{\mathbf{x} \in \mathcal{B}} \log p(\mathbf{x}; \Theta, \alpha)$
 9:   $\Theta \leftarrow \Theta - \eta \nabla_\Theta \mathcal{L}$
10:   Enforce $\pi$ on simplex (projection or softmax)
11:  **end for**
12: **end for**
13: **return** $\Theta$

---

**(iii) Integral convergence**: We have $0 \le w_k \le 1$ for all $\mathbf{u}, \alpha$. Pointwise limit gives $w_k \to g_k$ except on measure-zero boundaries. By dominated convergence:

$$\lim_{\alpha \to \infty} \int w_k(\mathbf{u}; \alpha) p(\mathbf{u}) \, d\mathbf{u} = \int g_k(\mathbf{u}) p(\mathbf{u}) \, d\mathbf{u} = \int_{V_k} p(\mathbf{u}) \, d\mathbf{u}.$$

$\square$

The margin $\gamma(\mathbf{u})$ quantifies how much closer $\mathbf{u}$ is to its nearest centroid compared to second-nearest. Large margins yield faster convergence. Small margins (near decision boundaries) require larger $\alpha$ for hard-like behavior.

## B. Synthetic Dataset Construction

Figure 5 visualizes the synthetic 2D/3D benchmarks used in our experiments. These datasets were designed to emphasize geometric structure (e.g., curved manifolds, crossings, knots, and disconnected supports) that can be difficult to capture with input-independent mixture weights, making them well-suited for evaluating geometry-aware routing. All datasets were generated following a unified protocol: (1) generate raw samples from a geometric structure as defined below, (2) add Gaussian noise $\mathcal{N}(0, \sigma^2\mathbf{I})$ with $\sigma = 0.01$, (3) standardize to zero mean and unit variance per dimension, (4) split into $10k/5k/5k$ train/val/test sets.

### B.1. Two-Dimensional Datasets

**CheckerBoard.** Nine Gaussian clusters at grid positions $\{-1.5, 0, 1.5\}^2$. Each cluster samples uniformly from a square $[\mathbf{c}_k - 0.6, \mathbf{c}_k + 0.6]$ then adds Gaussian noise $\sigma = 0.15$ and clips to bounds.

**Pinwheel.** Five radial arms at angles $2\pi k/5$ for $k = 0, \ldots, 4$. Each sample: radius $r \sim \mathcal{N}(1.0, 0.3^2)$, angular offset $\delta\theta \sim \mathcal{N}(0, 0.2^2)$, then convert to Cartesian $(r\cos(\theta_k + \delta\theta), r\sin(\theta_k + \delta\theta))$.

**Spiral.** Two interleaved Archimedean spirals. For each spiral: sample $\theta \sim \text{Uniform}(0, 2\pi)$, set $\theta \leftarrow \sqrt{\theta} \cdot 2\pi$ (arc-length correction), radius $r = 2\theta$, convert to $(\pm r\cos\theta, \pm r\sin\theta)$ (opposite signs for two spirals), add noise $\sigma = 0.1$.

**Alphabet.** Samples uniformly from pixel-based $7 \times 5$ binary grids representing uppercase letters arranged spatially. The results reported are for the letter $W$. Active pixels are converted to continuous coordinates (cell size 0.2), positioned in a grid layout with letter gap 0.4.

### B.2. Three-Dimensional Datasets

All 3D datasets were generated by sampling a parameter $t \sim \text{Uniform}([-\pi, \pi])$, apply a parametric curve as defined below, scaling by 4, adding noise $\sigma = 0.01$, foloowing (Sidheekh et al., 2022; 2023)

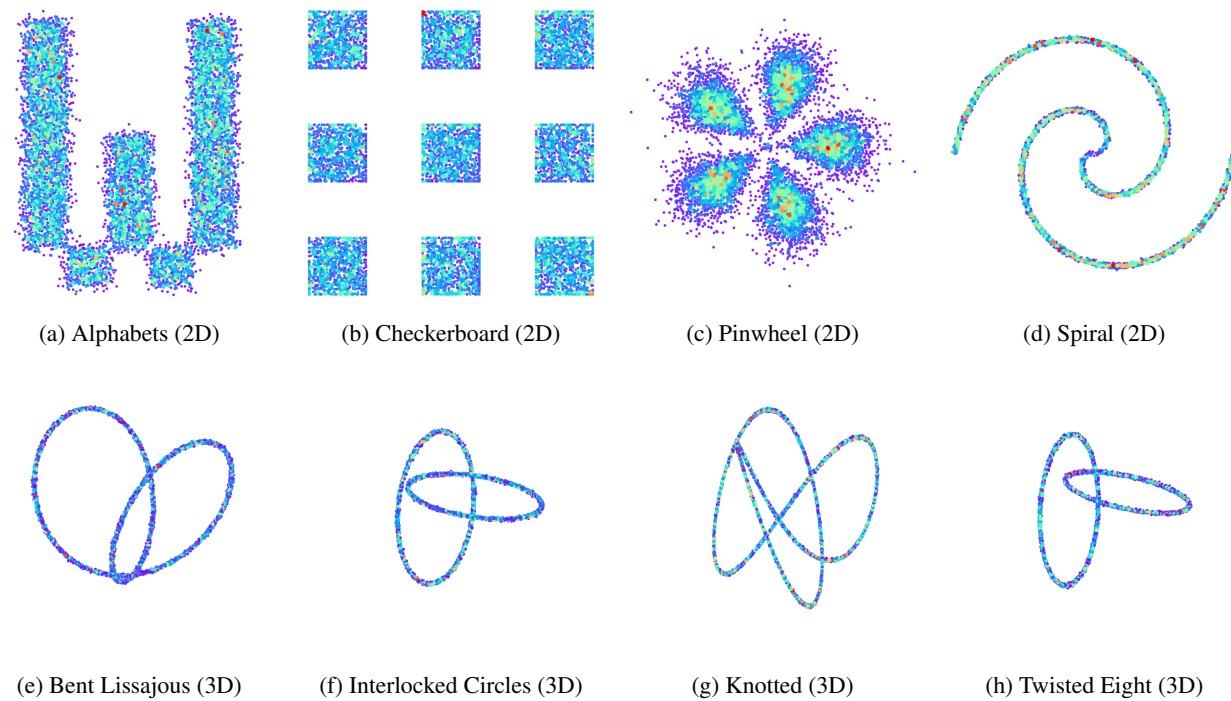

(a) Alphabets (2D)    (b) Checkerboard (2D)    (c) Pinwheel (2D)    (d) Spiral (2D)

(e) Bent Lissajous (3D)    (f) Interlocked Circles (3D)    (g) Knotted (3D)    (h) Twisted Eight (3D)

*Figure 5.* **Synthetic 2D/3D density estimation benchmarks.** Top row: 2D datasets with diverse local geometry and disconnected support. Bottom row: 3D manifold-like datasets exhibiting crossings, interlocks, and knotting. These benchmarks stress-test whether geometry-aware routing can specialize locally while maintaining reliable inference (VT via certification; HFV via alignment).

**BentLissajous.** Lissajous curve: $(x, y, z) = (\sin(2t), \cos(t), \cos(2t))$.

**InterlockedCircles.** Two circles in orthogonal planes: Circle 1 in $xy$-plane $(\sin t, \cos t, 0)$, Circle 2 in $xz$-plane $(1 + \sin t, 0, \cos t)$.

**Knotted.** Trefoil knot: $(x, y, z) = (\sin t + 2 \sin 2t, \cos t - 2 \cos 2t, \sin 3t)$.

**TwistedEight.** Two circles in orthogonal planes: $(\sin t, \cos t, 0)$ and $(2 + \sin t, 0, \cos t)$.

## C. Implementation Details

All models were implemented in PyTorch using the CirKit package (Lab, 2024) and trained with Adam (lr 0.01, batch size 500, 100 epochs) on a single 24GB NVIDIA L4 GPU. VT and HFV gating were implemented as drop-in replacements for sum layers, with learnable centroids initialized via $k$-means and optimized jointly with circuit parameters. We annealed the soft-gating inverse temperature linearly, $\alpha : 1 \rightarrow 50$, to transition from smooth routing to near-hard assignments. For VT models, evaluation and model selection used the certified normalization bounds produced by our box-based inference routine (reporting the log-likelihood lower bound), whereas HFV models preserved exact tractable inference throughout. For all models, the best performing epoch in terms of validation performance was saved and loaded for testing.

