# OpenReview forum: "Geometry-Aware Probabilistic Circuits via Voronoi Tessellations"
_ICML.cc/2026/Conference — ICML 2026 regular_

### Official Review · Reviewer_osHr · 2026-02-24

**Soundness:** 4
**Presentation:** 3
**Significance:** 4
**Originality:** 3
**Overall Recommendation:** 5
**Confidence:** 4

**Summary:**

This article studies a central concept in the theory of tractable probabilistic modeling:
How to enhance the expressivity of probabilistic circuits (PCs) without forfeiting their hallmark property of exact and efficient inference?

Naively integrating Voronoi-gated routing into smooth and decomposable PCs breaks the recursive factorization of integrals that underpins tractable inference.
Even with fully factorized experts, integrating over oblique Voronoi cells generally requires computing integrals over convex polytopes, which do not decompose along product-node scopes.
To address this, the authors propose two complementary solutions.
The first is a certified approximate inference framework that replaces Voronoi cells with inner and outer axis-aligned box approximations, enabling provable lower and upper bounds on partition functions, marginals, and conditionals.

The second is a structural solution: Hierarchical Factorized Voronoi (HFV) PCs, in which the Voronoi partition is aligned with the circuit’s variable decomposition (e.g., via a vtree).

**Compliance With Llm Reviewing Policy:**

Affirmed.

**Final Justification:**

The rebuttal addressed my main concerns.

**Key Questions For Authors:**

How does the certified VT framework behave in higher-dimensional settings (e.g., $> 20$ dimensions)?

Can the authors characterize (theoretically or empirically) the class of decision boundaries representable by HFV-PCs compared to general VT-PCs?
Are there concrete examples where HFV fails but VT succeeds?

In deeper PC architectures, does hierarchical Voronoi alignment meaningfully improve sample efficiency or interpretability compared to simply increasing mixture components?

**Limitations:**

Yes.

**Strengths And Weaknesses:**

Strengths

The formalization of the tension between geometric routing and decomposability is a strong conceptual contribution.
The intractability results (e.g., cell-restricted integrals over oblique polytopes) are clearly motivated and well integrated into the PC literature.

The dual strategy, that is, certified approximate inference for full Voronoi gating and structural alignment for exact HFV-PCs, is elegant.

The exponential convergence guarantee under a geometric margin adds theoretical depth and justifies annealing as a principled training strategy rather than a heuristic.

Weaknesses

HFV restores tractability by imposing alignment constraints, but this can significantly restrict geometric expressivity.
The empirical results could suggest that performance is similar to baselines in shallow regimes.

---

> ### Author Rebuttal · Authors · 2026-03-30
>
> We thank the reviewer for the careful and supportive review.
>
> ---
>
> **Q1: Behavior of the certified VT framework in higher dimensions.** Theorem 3.9 makes the dependence explicit: under uniform refinement, the gap decreases as $O(2^{-t/d})$. Thus, the curse of dimensionality is real for the unrestricted VT framework. Future extensions include studying clever ways to reduce this curse of dimensionality. In moderate dimensions we still expect adaptive refinement remains useful because it prioritizes boxes with the largest gap contribution, but beyond that regime HFV is the scalable exact alternative. We will clarify this more directly in the paper.
>
> ---
>
> **Q2: Expressivity of HFV versus general VT.** HFV restricts the geometry so that the gating structure factorizes along the circuit decomposition. This preserves tractability but limits the class of boundaries that can be represented directly. A simple example is data separated by an oblique boundary such as a $45^\circ$ rotated split: VT can represent such a partition directly, whereas HFV must approximate it through a hierarchical composition of factorized regions. This is analogous to the difference between oblique and axis-aligned decision trees. We agree that a more formal characterization of this representational gap would be valuable and interesting future work.
>
> ---
>
> **Q3: Why hierarchical alignment instead of simply using more mixture components?** The key advantage is both structural and interpretive, not merely parametric. HFV yields an explicit hierarchical spatial decomposition in which each cell corresponds to a region with a responsible expert. This can improve transparency, debugging, and interpretability in a way that simply increasing the number of mixture components does not guarantee. It can also lay the grounds for adapting PCs to changing data distributions in a continual learning setting, where the relevant experts can be pruned or retrained without retraining the entire model. Moreover, as shown recently [1] , simply scaling PCs via more mixture components does NOT seem to correspondingly help achieve better performance, as optimization typically tends to get stuck and there is a need for incorporating additional supervisory signal into the learning of sum nodes, and our geometric routing framework is one way to achieve this.
>
> [1]  Anji Liu, Honghua Zhang, and Guy Van den Broeck. "Scaling up probabilistic circuits by latent variable distillation." ICLR 23
>
> ---

---

> > ### Author Rebuttal · Reviewer_osHr · 2026-04-01
> >
> > My concerns have been adequately addressed.

---

> > > ### Author Response · Authors · 2026-04-06
> > >
> > > We thank the reviewer for the positive assessment and for confirming that the main concerns were adequately addressed.
> > >
> > > As an update on scalability to higher dimensions, we have used the time from the discussion phase to run some additional experiments on four standard UCI tabular density-estimation benchmarks, specifically: $\texttt{power} (6D), \texttt{gas} (8D), \texttt{hepmass} (21D)$, and $\texttt{miniboone} (43D)$, using both VT and HFV variants without refinement. These are standard density estimations benchmarks commonly used in the PC and normalizing flow literature [1,2]. For all models, we kept the base
> > > architecture the same with 40 input and sum units for a fair comparison. The results below report **mean test log-likelihood averaged across 3 random trials**; VT rows report certified lower (and upper) bounds.
> > >
> > > ---
> > >
> > >
> > >
> > >  $\hspace{8em} \textbf{EinsumNet Family}:$
> > >
> > >
> > > | Model | power | gas | hepmass | miniboone |
> > > |---|---|---|---|---|
> > > | EinsumNet | 0.558 | 5.851 | -22.348 | -32.242 |
> > > | HFV-EinsumNet | 0.557 | 5.863 | **-22.319** | **-32.179** |
> > > | VT-EinsumNet [lower] | **1.728** | **7.130** | -22.976 | -32.823 |
> > > | VT-EinsumNet [upper] | 3.463 | 9.762 | -4.815 | -14.073 |
> > >
> > >
> > >
> > >  $\hspace{8em} \textbf{HCLT Family}:$
> > >
> > > | Model | power | gas | hepmass | miniboone |
> > > |---|---|---|---|---|
> > > | HCLT | 0.553 | 7.220 | -20.356 | -25.428 |
> > > | HFV-HCLT | 0.549 | 7.068 | -20.354 | **-25.356** |
> > > | VT-HCLT [lower] | **5.096** | **10.572** | **-19.095** | -26.458 |
> > > | VT-HCLT [upper]  $\qquad $  | 6.766 | 13.165 | -1.608 | -8.954 |
> > > ---
> > >
> > > There are two observations that are worth noting here.
> > >
> > > First, **VT's certified lower bounds either exceed or closely match the exact likelihoods of the corresponding baselines on all datasets.**  In particular, VT-HCLT outperforms all baselines on power (+4.543), gas (+3.352), and hepmass (+1.261). VT-EinsumNet improves over its baseline on power (+1.170) and gas (+1.279). Since these are lower bounds, the true VT likelihood is expected to be even higher. This confirms that the geometric expressivity of Voronoi routing is applicable and yields real gains even beyond the 2D/3D synthetic setting.
> > >
> > > Second, **the gap between lower and upper bounds widens with dimension, as acknowledged and  predicted by our theory** (Theorem 3.9). This is also visible in the [learning curves](https://ibb.co/5gCbc4Tp): the certified interval $[Z^{-}, Z^{+}]$ is tight on $\texttt{power}$ and $\texttt{gas}$, and wider on $\texttt{hepmass}$ and $\texttt{miniboone}$. This behavior is expected from the current conservative inner-box construction, rather than indicating a limitation of the overall framework. Importantly, the lower bound itself, the quantity relevant for model comparison remains competitive or superior. As other reviewers and us have noted, tighter geometric constructions (e.g., approximate Voronoi diagrams, adaptive decompositions, or coordinate-wise optimization) fit naturally into our framework and should further reduce this gap, and are important future directions for us.
> > >
> > > We hope these additional experiments and results make it clearer that the framework extends meaningfully beyond toy settings, and strengthen the paper's main contribution of introducing a principled foundation for geometry-aware routing in PCs.
> > >
> > > Learning curves: https://ibb.co/5gCbc4Tp
> > >
> > >
> > > - [1] Papamakarios, George, Theo Pavlakou, and Iain Murray. Masked autoregressive flow for density estimation. Advances in Neural Information Processing Systems (NeurIPS), 2017.
> > > - [2] Loconte, Lorenzo, et al. Subtractive Mixture Models via Squaring. Proceedings of the International Conference on Learning Representations (ICLR), 2024.

---

### Official Review · Reviewer_aouH · 2026-03-14

**Soundness:** 2
**Presentation:** 3
**Significance:** 2
**Originality:** 3
**Overall Recommendation:** 4
**Confidence:** 2

**Summary:**

This paper introduces geometry-aware, input-dependent routing into probabilistic circuits using Voronoi tessellations. The paper first formalizes that naive Voronoi-gated sum nodes break tractability. It then proposes two complementary solutions: a certified approximate inference framework, and an exactly tractable Hierarchical Factorized Voronoi (HFV) construction. To enable end-to-end training, the paper introduces a soft Voronoi gating relaxation and proves soft-to-hard convergence. Experiments on several synthetic 2D/3D density-estimation tasks show that VT variants can improve likelihood lower bounds over baseline PCs.

**Compliance With Llm Reviewing Policy:**

Affirmed.

**Final Justification:**

My major concerns are resolved in the rebuttal. I will increase my current rating.

**Key Questions For Authors:**

1. The paper proposes two complementary routes, VT with certified approximate inference and HFV with exact tractability. In practice, when should a user prefer one over the other?

2. Can the authors provide stronger empirical evidence on the tightness of the bounds, the cost-quality tradeoff of refinement, and how this behaves as dimensionality grows?

3. The experiments are positioned on synthetic data. Can the authors add evaluation on more realistic continuous density-estimation tasks or higher-dimensional benchmarks to better support the method?

4. Can the authors provide more training details and sensitivity analysis for the annealing schedule and centroid initialization?

**Limitations:**

No. The paper mentions some limitations implicitly, but the discussion is not strong enough. Please refer to the weaknesses.

**Strengths And Weaknesses:**

Strengths

1. The paper addresses an interesting and meaningful problem. A core limitation of standard probabilistic circuits is that they are typically fixed and data-independent, which makes them less effective at adapting to local geometric structure in the input space. The paper proposes geometry-aware routing through Voronoi tessellations as a way to make PCs more adaptive, which is a promising direction.

2. The paper is well structured. The author explains why naive Voronoi gating breaks tractable inference, and develops two separate solutions to this issue. This makes the paper feel conceptually organized and technically serious.

Weaknesses

1. The empirical evaluation is too limited to fully support the paper’s claims. The experiments are restricted to a small set of synthetic 2D and 3D density-estimation tasks, with limited baselines and only a few runs. That is enough for proof of concept, but not enough to demonstrate that the method matters in realistic PC applications or scales to more challenging data. I would expect stronger evidence on more real-world benchmarks.

2. The practical improvements of the proposed methods remains somewhat unclear. The author claims unconstrained VT variant appears more expressive, but then only approximate inference is available, and the experiments do not yet establish how tight or useful those approximations are in practice. The reported likelihood improvements for HFV over the baseline are often modest. As a result, the paper does not yet clearly show a decisive practice of which methods is more attractive.

3. The broader impact remains somewhat speculative. The framework is interesting from a conceptual standpoint, but the current empirical evidence does not yet show that geometry-aware PCs are broadly useful in realistic downstream tasks.

---

> ### Author Rebuttal · Authors · 2026-03-30
>
> We thank the reviewer for the encouraging assessment and the concrete suggestions for strengthening the paper.
>
> **W1: Limited empirical evaluation / Scalability and high-dimensional settings.** We agree that the current experiments are proof-of-concept and limited to low-dimensional synthetic settings rather than comprehensive benchmarking. This is already stated in the submission, and we will make that framing more prominent. Our primary aim in this paper is to establish a theoretical foundation for geometry-aware routing in PCs. The empirical section is intended to validate that the constructions behave as expected in practice, not yet to claim broad superiority across real-world benchmarks.
>
> At the same time, we would like to emphasize we still make significant *foundational and theoretical* contributions in this work. In particular, we:  (i) formally characterize why even simple geometric routing breaks tractable inference, which is an issue not analyzed in prior work on data-dependent mixtures;   (ii) develop a certified inference framework with lower and upper bounds and *anytime refinement*, and show how these guarantees propagate to partition functions, marginals, and conditionals; and  (iii) identify a structural condition (HFV) that restores exact tractability, together with a soft-to-hard convergence result with exponential rate under a geometric margin.
>
> These results are dimension-independent as mathematical statements, even though the practicality of the current (deliberately simple) VT instantiation degrades with dimension.
>
> Our intent is therefore not to present VT as a final scalable solution, but as a framework that establishes that *certified approximate inference is possible in the more expressive geometric setting*, namely unconstrained Voronoi routing over the modeled space. This enables more expressive PCs while retaining correctness guarantees, and provides a foundation for more sophisticated geometric approximations (e.g., adaptive or learned decompositions) that could better address the curse of dimensionality.
>
> The two proposed approaches also serve complementary roles. VT provides expressivity together with certified approximate inference, but is subject to the curse of dimensionality under naive refinement. HFV, in contrast, is designed as the *scalable exact alternative*. By aligning geometric routing with the circuit decomposition, it avoids box refinement entirely and restores exact tractable inference in a manner comparable to standard PCs. In addition, our HFV formulation also enables **tensorized deterministic PCs** to be trained efficiently with gradient-based optimizers, and scaling them to very large models (e.g., millions to billions of parameters), improving practicality relative to prior approaches such as cutset networks.
>
> We do not claim to fully resolve scalability for unrestricted geometric routing in this work. Rather, we introduce and formalize this new direction: VT demonstrates feasibility with certified guarantees, while HFV provides a principled path to scalability. Developing more advanced approximation schemes that mitigate the curse of dimensionality is a natural and important direction for future work enabled by our framework.
>
> ---
>
> **W2: Practical gains and the role of VT vs. HFV.**  *HFV-PC* is the preferred choice when exact inference is required and the geometric structure can be represented well by hierarchical factorized partitions. Note that in higher dimensions, one can also restrict the HFV partitioning only to a subset of the PC layers, to avoid being fully deterministic.  *VT-PC* is preferable when one is willing to trade exactness for added expressive ability, while still retaining certified approximate inference, and would be useful when the target geometry contains oblique or cross-variable boundaries that HFV cannot represent directly.
>
> So in that sense, the two methods occupy complementary points in an expressivity–tractability tradeoff. We will make this tradeoff more explicit in the revision.
>
> ---
>
> **Q2: Tightness of bounds and dimensional behavior.** Theorem 3.9 provides the formal convergence guarantee and makes clear how the refinement burden scales with dimension. The paper also explicitly discusses the curse of dimensionality in Remark A.10. We agree that a more systematic study across dimensions would be valuable and will add this to the discussion of future work.
>
> ---
>
> **Q3: Realistic benchmarks.** We agree and view this as an important next step. Our present focus was on controlled settings where the geometric structure can be inspected and our approach validated.
>
> ---
>
> **Q4: Training details.** The current appendix already includes the main hyperparameter and training details: optimizer, learning rate, batch size, number of epochs, annealing schedule, centroid initialization, and architectural setup. If there are specific implementation details the reviewer would like to know, we would be happy to provide them as well.

---

> > ### Author Rebuttal · Reviewer_aouH · 2026-04-03
> >
> > I appreciate the authors' explanation on training details and tightness of the bound. However, my concerns still remain on the scalability of the VT to realistic benchmarks and high-dimensional data. The pracitcalness on such cases is not clear within the existing discussion. Therefore, I would like to maintain my current score.

---

> > > ### Author Response · Authors · 2026-04-06
> > >
> > > We thank the reviewer for the feedback, and we understand the remaining concern about scalability and applicability to realistic data. To address this,  we have used the time from the discussion phase to run some additional experiments on four standard UCI tabular density-estimation benchmarks in higher dimensions (D), specifically: $\texttt{power} (6D), \texttt{gas} (8D), \texttt{hepmass} (21D)$, and $\texttt{miniboone} (43D)$, using both VT and HFV variants without refinement. These are standard density estimations benchmarks commonly used in the PC and normalizing flow literature [1,2]. For all models, we kept the base
> > > architecture the same with 40 input and sum units for a fair comparison. The results below report **mean test log-likelihood averaged across 3 random trials**; VT rows report certified lower (and upper) bounds.
> > >
> > > ---
> > >
> > >
> > >
> > >  $\hspace{8em} \textbf{EinsumNet Family}:$
> > >
> > >
> > > | Model | power | gas | hepmass | miniboone |
> > > |---|---|---|---|---|
> > > | EinsumNet | 0.558 | 5.851 | -22.348 | -32.242 |
> > > | HFV-EinsumNet | 0.557 | 5.863 | **-22.319** | **-32.179** |
> > > | VT-EinsumNet [lower] | **1.728** | **7.130** | -22.976 | -32.823 |
> > > | VT-EinsumNet [upper] | 3.463 | 9.762 | -4.815 | -14.073 |
> > >
> > >
> > >
> > >  $\hspace{8em} \textbf{HCLT Family}:$
> > >
> > > | Model | power | gas | hepmass | miniboone |
> > > |---|---|---|---|---|
> > > | HCLT | 0.553 | 7.220 | -20.356 | -25.428 |
> > > | HFV-HCLT | 0.549 | 7.068 | -20.354 | **-25.356** |
> > > | VT-HCLT [lower] | **5.096** | **10.572** | **-19.095** | -26.458 |
> > > | VT-HCLT [upper]  $\qquad $  | 6.766 | 13.165 | -1.608 | -8.954 |
> > > ---
> > >
> > > There are two observations that are worth noting here.
> > >
> > > First, **VT's certified lower bounds either exceed or closely match the exact likelihoods of the corresponding baselines on all datasets.**  In particular, VT-HCLT outperforms all baselines on power (+4.543), gas (+3.352), and hepmass (+1.261). VT-EinsumNet improves over its baseline on power (+1.170) and gas (+1.279). Since these are lower bounds, the true VT likelihood is expected to be even higher. This confirms that the geometric expressivity of Voronoi routing is applicable and yields real gains even beyond the 2D/3D synthetic setting.
> > >
> > > Second, **the gap between lower and upper bounds widens with dimension, as acknowledged and  predicted by our theory** (Theorem 3.9). This is also visible in the [learning curves](https://ibb.co/5gCbc4Tp): the certified interval $[Z^{-}, Z^{+}]$ is tight on $\texttt{power}$ and $\texttt{gas}$, and wider on $\texttt{hepmass}$ and $\texttt{miniboone}$. This behavior is expected from the current conservative inner-box construction, rather than indicating a limitation of the overall framework. Importantly, the lower bound itself, the quantity relevant for model comparison remains competitive or superior. As both the reviewer and us have noted, tighter geometric constructions (e.g., approximate Voronoi diagrams, adaptive decompositions, or coordinate-wise optimization) fit naturally into our framework and should further reduce this gap, and are important future directions for us.
> > >
> > > We hope these additional experiments and results make it clearer that the framework extends meaningfully beyond toy settings, and strengthen the paper's main contribution in introducing a principled foundation for geometry-aware routing in PCs.
> > >
> > > Learning curves: https://ibb.co/5gCbc4Tp
> > >
> > > - [1] Papamakarios, George, Theo Pavlakou, and Iain Murray. Masked autoregressive flow for density estimation. Advances in Neural Information Processing Systems (NeurIPS), 2017.
> > > - [2] Loconte, Lorenzo, et al. Subtractive Mixture Models via Squaring. Proceedings of the International Conference on Learning Representations (ICLR), 2024.

---

### Official Review · Reviewer_t3xK · 2026-03-15

**Soundness:** 3
**Presentation:** 2
**Significance:** 3
**Originality:** 3
**Overall Recommendation:** 4
**Confidence:** 4

**Summary:**

This work proposes a new structural property for the sum node in the probabilsitic cicuit models called Voronoi-gated sum node. While the proposed property might lead to intractabilty in integral computation, it further proposes to approximate the Voronoi cells with axis-aligned boxes such that the integrals inside the boxes are tractable. It further proposes anytime bound refinement for bisecting the boxes to improve approximation quality. Learning algorithms for the proposed PCs and empirical evaluations on synthetic datasets are further presented.

**Compliance With Llm Reviewing Policy:**

Affirmed.

**Final Justification:**

I would like to thank the authors for the additional UCI dataset experiments which are helpful. Thus I increase my score from 3 to 4.

**Key Questions For Authors:**

- Why are the proposed Voronoi-gated sum node geometry-aware and input-dependent?
- How are the centroids defined or are they learned from the data?
- Can the authors comment about the scalability of the proposed PCs?

**Limitations:**

It seems this paper doesn't include an impact statement.

**Strengths And Weaknesses:**

The problem studied in this work on how to improve the PC modeling while maintaining tractability is interesting. I find Figure 2 helpful for understanding the proposed algorithms. However, a few concerns arise:
- It is unclear to me how the proposed Voronoi-gated sum nodes are related to the goal of enabling PCs to capture and adapt to the local geometric structure in the data manifold as stated in the introduction. While they are described as geometry-aware and input-dependent, they seem to be a specific class of deterministic PCs and it is unclear under what context they are preferred.
- The box approximations seem impractical to scale since with too few boxes, the approximations are too loose while the number of boxes required for decent approximations might explode. The experiment settings are toyish since they are over 2D and 3D synthetic data. There is no theoretical or empirical evidence to show whether the proposed algorithms scale to high-dimensional settings such as image datasets.
- The presentation of the work could be significantly improved. The paper is quite heavy on technical details, and it would be helpful to include a minimal working example to illustrate the proposed algorithms and make the overall pipeline easier to follow. In addition, Proposition 3.2 does not appear to provide a rigorous proof of intractability. The argument only considers the case of fully factorized distributions, where the integral is still tractable, and does not clearly establish why the problem becomes intractable in the general case.

---

> ### Author Rebuttal · Authors · 2026-03-30
>
> We thank the reviewer for the constructive comments and for identifying places where the paper can be made clearer.
>
> ---
>
> **W1 / Q1: Why are Voronoi-gated sum nodes geometry-aware and input-dependent?** Standard PC sum nodes use fixed mixture weights $\pi_k$, so the routing behavior is global and does not depend on where an input lies in the modeled space. In our Voronoi-gated node, the active child is determined by whether the input belongs to a Voronoi cell $V_k$, i.e., by proximity to learned centroids. Thus, different regions of the input space activate different experts. This is why the routing is both input-dependent and geometry-aware: it is explicitly induced by spatial structure in the modeled variables.
>
> Note that in practice, the gating need not happen in all the layers from the root to the leaves, in which case the PC would not be deterministic. Also, while we performed VT in the input space for ease of theoretical analysis, a natural future extension is studying VT and tractability of inference in a learned embedding space that can better capture the geometry on low dimensional manifolds.
>
> ---
>
> **Q2: How are the centroids defined?** The centroids are initialized by k-means and then learned jointly with the circuit parameters using the soft gating relaxation introduced in Section 3.3. During training, the gate is differentiable, which allows gradient-based updates to both the centroids and the circuit parameters. At test time, we revert to hard Voronoi assignments.
>
> ---
>
> **W2 / Q3: Scalability and high-dimensional settings.** We agree that the current experiments are proof-of-concept and limited to low-dimensional synthetic settings, and we will make this positioning more explicit. We also agree that naive unrestricted VT with box refinement is not scalable in high dimensions, and this limitation is explicitly acknowledged in the paper.
>
> Our goal in this work is to establish a *foundational framework*:  (i) to formally characterize why geometric routing breaks tractable inference,  (ii) to show that certified approximate inference is still possible via VT, and   (iii) to identify HFV as a structural condition that restores exact tractability.
>
> These results are dimension-independent as mathematical statements, even though the current VT instantiation degrades with dimension. Importantly, VT and HFV serve complementary roles: VT provides expressivity with certified approximate inference, while HFV provides a scalable exact alternative by avoiding refinement.
>
> **Please see W1 in our response to Reviewer aouH where we elaborate and discuss this issue in detail.**
>
> ---
>
> **W3: Proposition 3.2 and the notion of intractability.** We believe the reviewer's concern may stem from a misunderstanding of the role of Proposition 3.2. Tractable inference for marginals and conditionals in PCs is achieved via decomposability, this is both necessary and sufficient. Decomposability works by allowing joint integrals to factor into products of independent lower-dimensional integrals via Fubini's theorem, and this factorization is exactly what makes standard PCs tractable.
>
> What Proposition 3.2 shows is that even in the **simplest possible case**, i.e. a single sum node with fully factorized expert distributions, introducing Voronoi gating breaks this factorization. The integration domain $V_k$ is a convex polytope with oblique boundaries that couples variables the circuit tries to separate, hence breaks decomposability, thus making marginal/conditional inference intractable.
>
> The logic here is pretty straightforward: to establish a negative result (of intractability), it is **sufficient** to show that the simplest case is already intractable. If even fully factorized experts cannot be integrated tractably over Voronoi cells, then more complex (non-factorized) experts certainly cannot either, and the intractability of the general case follows logically. The proposition is not claiming that only the factored case is problematic; rather, it identifies the minimal setting in which the obstruction already appears, from which the intractability of more general cases follows immediately.
>
> We *believe the concrete counterexample given in the appendix makes this explicit*. For two centroids $c_1=(0,1)$ and $c_2=(1,0)$, the corresponding Voronoi cell is $\{(x_1,x_2): x_2 > x_1\}$. Integrating a factorized density over this region yields an iterated integral whose inner limit depends on the outer variable, so the recursive factorization used in PCs breaks. We have additionally cited classical hardness results for polytope volume computation to motivate why this issue is not merely superficial. In the revision, we will clarify that Proposition 3.2 establishes structural non-factorizability, not a standalone complexity claim.
>
> ---

---

> > ### Author Rebuttal · Reviewer_t3xK · 2026-04-03
> >
> > I would like to thank the authors for the clarifications. Still, my concern on the scalability of the proposed PCs remains and thus I keep my score.

---

> > > ### Author Response · Authors · 2026-04-06
> > >
> > > We thank the reviewer for the acknowledgement, and we understand that the key remaining concern is about scalability. We have used the time from the discussion phase to run some additional experiments on four standard UCI tabular density-estimation benchmarks in higher dimensions (D), specifically: $\texttt{power} (6D), \texttt{gas} (8D), \texttt{hepmass} (21D)$, and $\texttt{miniboone} (43D)$, using both VT and HFV variants without refinement. These are standard density estimations benchmarks commonly used in the PC and normalizing flow literature [1,2]. For all models, we kept the base
> > > architecture the same with 40 input and sum units for a fair comparison. The results below report **mean test log-likelihood averaged across 3 random trials**; VT rows report certified lower (and upper) bounds.
> > >
> > > ---
> > >
> > >
> > >
> > >  $\hspace{8em} \textbf{EinsumNet Family}:$
> > >
> > >
> > > | Model | power | gas | hepmass | miniboone |
> > > |---|---|---|---|---|
> > > | EinsumNet | 0.558 | 5.851 | -22.348 | -32.242 |
> > > | HFV-EinsumNet | 0.557 | 5.863 | **-22.319** | **-32.179** |
> > > | VT-EinsumNet [lower] | **1.728** | **7.130** | -22.976 | -32.823 |
> > > | VT-EinsumNet [upper] | 3.463 | 9.762 | -4.815 | -14.073 |
> > >
> > >
> > >
> > >  $\hspace{8em} \textbf{HCLT Family}:$
> > >
> > > | Model | power | gas | hepmass | miniboone |
> > > |---|---|---|---|---|
> > > | HCLT | 0.553 | 7.220 | -20.356 | -25.428 |
> > > | HFV-HCLT | 0.549 | 7.068 | -20.354 | **-25.356** |
> > > | VT-HCLT [lower] | **5.096** | **10.572** | **-19.095** | -26.458 |
> > > | VT-HCLT [upper]  $\qquad $  | 6.766 | 13.165 | -1.608 | -8.954 |
> > > ---
> > >
> > > There are two observations that are worth noting here.
> > >
> > > First, **VT's certified lower bounds either exceed or closely match the exact likelihoods of the corresponding baselines on all datasets.**  In particular, VT-HCLT outperforms all baselines on power (+4.543), gas (+3.352), and hepmass (+1.261). VT-EinsumNet improves over its baseline on power (+1.170) and gas (+1.279). Since these are lower bounds, the true VT likelihood is expected to be even higher. This confirms that the geometric expressivity of Voronoi routing is applicable and yields real gains even beyond the 2D/3D synthetic setting.
> > >
> > > Second, **the gap between lower and upper bounds widens with dimension, as acknowledged and  predicted by our theory** (Theorem 3.9). This is also visible in the [learning curves](https://ibb.co/5gCbc4Tp): the certified interval $[Z^{-}, Z^{+}]$ is tight on $\texttt{power}$ and $\texttt{gas}$, and wider on $\texttt{hepmass}$ and $\texttt{miniboone}$. This behavior is expected from the current conservative inner-box construction, rather than indicating a limitation of the overall framework. Importantly, the lower bound itself, the quantity relevant for model comparison remains competitive or superior. As discussed in our earlier response, and as pointed out by other reviewers, and in the paper, there are concrete paths to tightening these bounds: such as coordinate-wise radius optimization, approximate Voronoi diagrams, adaptive decompositions, etc. All of these fit naturally into our certified bound-propagation framework and are important future directions we aim to pursue.
> > >
> > > We would like to reiterate that while the point of this paper is to lay the theoretical foundations for geometry-aware routing in PCs: formalizing what breaks, why it breaks, and how to fix it,  introduce a (deliberately simple) way to achieve certified approximate inference, and a structural condition (HFV) that restores exact tractability, the additional results above make it clearer that the framework extends meaningfully beyond toy settings and is practically useful on standard benchmarks as well. More sophisticated refinement strategies to improve the bounds and scalability are venues for future work that can be built on this foundation.
> > >
> > > Learning curves: https://ibb.co/5gCbc4Tp
> > >
> > > - [1] Papamakarios, George, Theo Pavlakou, and Iain Murray. Masked autoregressive flow for density estimation. Advances in Neural Information Processing Systems (NeurIPS), 2017.
> > > - [2] Loconte, Lorenzo, et al. Subtractive Mixture Models via Squaring. Proceedings of the International Conference on Learning Representations (ICLR), 2024.

---

### Official Review · Reviewer_wM4D · 2026-03-16

**Soundness:** 2
**Presentation:** 3
**Significance:** 2
**Originality:** 3
**Overall Recommendation:** 4
**Confidence:** 3

**Summary:**

Probabilistic circuits are attractive generative models because they support exact and efficient inference through structural properties such as decomposability and smoothness. However, standard PCs typically use mixture weights that do not depend on the input, which can limit their ability to capture local geometric structure in the data. This paper explores how to introduce geometry-aware routing into PCs using Voronoi partitions, where different experts are responsible for different regions of the input space. The authors argue that this creates a tension with tractable inference, since Voronoi regions couple variables and break the factorization properties PCs rely on.

To address this challenge, the paper proposes two models. The first model, VT-PC, introduces Voronoi-gated sum nodes and performs approximate inference by computing certified lower and upper bounds on integrals over Voronoi cells. The key idea is to approximate each Voronoi cell with axis-aligned boxes so that integration remains tractable within the PC structure. These bounds are propagated through the circuit and refined through an iterative spatial subdivision procedure that gradually tightens the approximation. The second model, HFV-PC, restricts the Voronoi structure so that routing decisions align with the hierarchical decomposition of variables in the circuit, allowing the model to recover exact tractable inference while still benefiting from geometry-aware routing.

The empirical evaluation focuses on density estimation on several synthetic datasets with strong geometric structure, such as spirals, checkerboards, and knotted manifolds in two and three dimensions. The proposed models are compared against baseline PCs including EinsumNet and HCLT. The results show that geometry-aware PCs, particularly HFV-PC, achieve consistently higher test log-likelihood and better capture the geometric patterns present in the data. The experiments also illustrate how the bounding procedure in VT-PC converges as the spatial refinement increases.

**Compliance With Llm Reviewing Policy:**

Affirmed.

**Final Justification:**

The authors have addressed most of my concerns, and because of that, I increased my score from 3 to 4.

**Key Questions For Authors:**

* The inner-box and outer-box constructions appear very conservative. Have the authors explored tighter initial approximations or coordinate-wise optimization to improve the bounds? For instance, the notion of approximate Voronoi diagrams uses axis-aligned partitioning of space with n/eps^d boxes, where there are smaller boxes close to the boundaries. While this approach introduces an additional eps^{-d} term, it removes the need for the refinement step (which seems to have a similar complexity at the end of the day).
* During refinement, how do the authors determine whether a box intersects a Voronoi cell? Corner tests alone are insufficient in general, but it seems like the box classification section and remark A.12 suggest using corner testing to check box intersections.
* Have the authors considered alternative geometric approximations of Voronoi cells (e.g., adaptive spatial decompositions or approximate Voronoi diagrams)?
* Have the authors conducted experiments on higher-dimensional or real-world datasets?

**Limitations:**

Yes

**Strengths And Weaknesses:**

### Strengths
* The paper explores an interesting and relatively underexplored direction: incorporating geometric structure into probabilistic circuits.
* The paper identifies and analyzes a real tension between geometry-based routing and the tractable inference guarantees that make PCs attractive. The VT-PC framework provides certified bounds for inference, which is theoretically appealing.
* The HFV-PC construction is a clever idea that restores exact inference through structural alignment between routing and circuit scopes.

### Weaknesses
* The positioning of the paper in the literature somewhat overstates the novelty of introducing input-dependent routing in probabilistic circuits. Prior work such as conditional SPNs, probabilistic neural circuits, and related extensions already introduce mechanisms where mixture weights depend on the input or context. While the paper does cite some of these works, the introduction frames the limitation of standard PCs primarily as the lack of data-dependent routing. The actual novelty of the paper appears to lie more specifically in introducing Voronoi-based geometric routing together with tractability analysis, rather than in adaptive routing itself. Clarifying this distinction would improve the accuracy of the paper’s positioning relative to existing literature.
* The geometric approximation strategy is relatively simplistic compared to known computational geometry approaches such as approximate Voronoi diagrams or adaptive box decompositions.
* The experiments are limited to synthetic datasets with strong geometric structure, which naturally favor the proposed method.
* All datasets are very low dimensional (2D or 3D), leaving open the question of whether the approach scales to higher-dimensional problems.

---

> ### Author Rebuttal · Authors · 2026-03-30
>
> We thank the reviewer for the thoughtful review and for highlighting the promise and merits of the approach and the suggestion to clarify its positioning.
>
> ---
>
> **W1: Positioning in the literature.** We agree that the current introduction can better distinguish our contribution from prior forms of data dependent mixtures in PCs. Our goal is not to claim novelty for *input-dependent mixtures* in PCs in general. Rather, the novelty of this work is the introduction of *geometric routing via Voronoi tessellations* inside PCs, together with a rigorous analysis of the tractability issues this creates, and two complementary solutions:
>
> (1) A certified approximate inference framework for general Voronoi-gated PCs
> (2)  A structural condition, HFV, under which exact tractability is recovered.
>
> This distinction is central to the paper and we will sharpen it in the revision.
>
> As briefly discussed in Section 2 of the paper, works such as CSPNs, SPQNs, PNCs, and related models also introduce forms of context or input dependent sum nodes. However our claims and contribution in this work specifically is that we study what happens when routing is determined by *explicit geometric regions over the modeled variables themselves*, instantiated here through Voronoi cells. This difference is technically significant. CSPNs use neural networks to condition on external observed features, so they lose tractability of inference over the conditioned variables. The routing happening there is therefore fixed once those external features are given and does not induce the polytope-integration issue we study. SPQNs and PNCs rely on graphs- or ordering-based dependencies rather than geometric partitions of the modeled space, and as a result they trade away the any-order tractable marginalization that standard PCs provide.
>
> In contrast, Voronoi routing partitions the modeled variables into convex polyhedral regions, which creates a new and unique, yet solvable inference challenge: one must integrate factorized densities over oblique polytopes. We will revise the introduction to make this contribution more precise and avoid overstating novelty.
>
> ---
>
> **W2: Simplicity of the box approximation framework.** We agree that the current inner/outer box constructions are deliberately simple. Our intent here was to establish the framework and its guarantees as clearly as possible. The core contribution is that certified bounds can be constructed and refined monotonically, with convergence guarantees. More sophisticated geometric approximations, including adaptive decompositions, approximate Voronoi diagrams, and kd-tree style methods, are fully compatible with our framework and can be viewed as tighter instantiations of the same general principle. We will discuss this more explicitly in the revision.
>
> Regarding the comparison to approximate Voronoi diagrams: we view our refinement procedure as similar in spirit, but with an important practical distinction. Our method is *anytime*: at every refinement stage it returns valid certified bounds, and one may stop as soon as the gap is sufficiently small.
>
> ---
>
> **Q1: Tighter initial approximations.** Yes, tighter inner boxes can be obtained by optimizing coordinate-wise radii subject to the half-space constraints defining the cell. We mention this possibility after Proposition 3.8, but chose the conservative closed-form construction for simplicity and clarity. We will make this point more explicit.
>
> ---
>
> **Q2: How box-cell intersection is determined.** We do not rely on corner tests alone. The main procedure uses the half-space representation of the Voronoi cell. For a box   $B = \prod_i [\ell_i, u_i]$   and a half-space with normal vector $a$, we compute:  $\max_{x \in B} a^\top x$   and  $\min_{x \in B} a^\top x$ by selecting $u_i$ or $\ell_i$ depending on the sign of $a_i$. This yields an exact test for containment and emptiness with respect to convex polyhedral cells. The corner-based remark in the appendix was meant to convey the curse of dimensionality associated with the brute-force characterization, and the need for developing more clever methods. We will clarify this distinction in the main text.
>
> ---
>
> **Q3: Alternative geometric approximations.** Yes, we **have considered these as natural future extensions**. Adaptive spatial decompositions, approximate Voronoi diagrams, and related geometric data structures would likely improve practical tightness, while fitting naturally into the same certified bound-propagation framework. But **we emphasize that the scope here is to** lay the foundations and introduce the idea of Voronoi tessellations within probabilistic circuits (for the first time).
>
> ---
>
> **Q4: Higher-dimensional or real-world experiments.** Not yet, we agree this is an important direction for follow-up work. Our current submission focuses on establishing the theoretical foundation and validating the approach in controlled geometric settings. (Please see W1 in our response to Reviewer aouH)

---

> > ### Author Rebuttal · Reviewer_wM4D · 2026-04-04
> >
> > I thank the authors for their clarification. Most of my questions have been addressed, and I have increased my score to 4.
> >
> > That said, the scalability of the approach and its applicability to real-world datasets remain quite limited, as also noted by other reviewers and acknowledged by the authors. In its current form, the paper is best viewed as "to establish a theoretical foundation for geometry-aware routing in PCs" (quoted from authors' response), which is still a valuable contribution.
> >
> > I will not increase my score further, as I do not see a clear path forward that addresses the scalability limitations.

---

> > > ### Author Response · Authors · 2026-04-06
> > >
> > > We thank the reviewer for the updated assessment and for increasing the score. We appreciate the recognition that the paper provides a valuable theoretical foundation.
> > >
> > > Regarding the remaining concern about scalability, we have used the time from the discussion phase to run some additional experiments on four standard UCI tabular density-estimation benchmarks in higher dimensions (D), specifically: $\texttt{power} (6D), \texttt{gas} (8D), \texttt{hepmass} (21D)$, and $\texttt{miniboone} (43D)$, using both VT and HFV variants without refinement. These are standard density estimations benchmarks commonly used in the PC and normalizing flow literature [1,2]. For all models, we kept the base
> > > architecture the same with 40 input and sum units for a fair comparison. The results below report **mean test log-likelihood averaged across 3 random trials**; VT rows report certified lower (and upper) bounds.
> > >
> > > ---
> > >
> > >
> > >
> > >  $\hspace{8em} \textbf{EinsumNet Family}:$
> > >
> > >
> > > | Model | power | gas | hepmass | miniboone |
> > > |---|---|---|---|---|
> > > | EinsumNet | 0.558 | 5.851 | -22.348 | -32.242 |
> > > | HFV-EinsumNet | 0.557 | 5.863 | **-22.319** | **-32.179** |
> > > | VT-EinsumNet [lower] | **1.728** | **7.130** | -22.976 | -32.823 |
> > > | VT-EinsumNet [upper] | 3.463 | 9.762 | -4.815 | -14.073 |
> > >
> > >
> > >
> > >  $\hspace{8em} \textbf{HCLT Family}:$
> > >
> > > | Model | power | gas | hepmass | miniboone |
> > > |---|---|---|---|---|
> > > | HCLT | 0.553 | 7.220 | -20.356 | -25.428 |
> > > | HFV-HCLT | 0.549 | 7.068 | -20.354 | **-25.356** |
> > > | VT-HCLT [lower] | **5.096** | **10.572** | **-19.095** | -26.458 |
> > > | VT-HCLT [upper]  $\qquad $  | 6.766 | 13.165 | -1.608 | -8.954 |
> > > ---
> > >
> > > There are two observations that are worth noting here.
> > >
> > > First, **VT's certified lower bounds either exceed or closely match the exact likelihoods of the corresponding baselines on all datasets.**  In particular, VT-HCLT outperforms all baselines on power (+4.543), gas (+3.352), and hepmass (+1.261). VT-EinsumNet improves over its baseline on power (+1.170) and gas (+1.279). Since these are lower bounds, the true VT likelihood is expected to be even higher. This confirms that the geometric expressivity of Voronoi routing is applicable and yields real gains even beyond the 2D/3D synthetic setting.
> > >
> > > Second, **the gap between lower and upper bounds widens with dimension, as acknowledged and  predicted by our theory** (Theorem 3.9). This is also visible in the [learning curves](https://ibb.co/5gCbc4Tp): the certified interval $[Z^{-}, Z^{+}]$ is tight on $\texttt{power}$ and $\texttt{gas}$, and wider on $\texttt{hepmass}$ and $\texttt{miniboone}$. This behavior is expected from the current conservative inner-box construction, rather than indicating a limitation of the overall framework. Importantly, the lower bound itself, the quantity relevant for model comparison remains competitive or superior. As both the reviewer and us have noted, tighter geometric constructions (e.g., approximate Voronoi diagrams, adaptive decompositions, or coordinate-wise optimization) fit naturally into our framework and should further reduce this gap, and are important future directions for us.
> > >
> > > We hope these additional experiments and results make it clearer that the framework extends meaningfully beyond toy settings, while the paper's main contribution remains introducing a principled foundation for geometry-aware routing in PCs.
> > >
> > > Learning curves: https://ibb.co/5gCbc4Tp
> > >
> > > - [1] Papamakarios, George, Theo Pavlakou, and Iain Murray. Masked autoregressive flow for density estimation. Advances in Neural Information Processing Systems (NeurIPS), 2017.
> > > - [2] Loconte, Lorenzo, et al. Subtractive Mixture Models via Squaring. Proceedings of the International Conference on Learning Representations (ICLR), 2024.

---

### Decision · Program_Chairs · 2026-04-30

**Decision:**

Accept (regular)

**Comment:**

This paper introduces input-aware routing into probabilistic circuits (PCs) using Voronoi tessellations. However, naively gating sum nodes with Voronoi regions breaks exact tractable inference by requiring integration over convex polytopes. To resolve this, the paper proposes two complementary approaches: an approximate inference framework that uses spatial subdivision to provide certified lower and upper bounds, and a structural condition (Hierarchical Factorized Voronoi) that aligns routing with the circuit’s variable decomposition to recover exact tractable inference.

The main shared criticism among reviewers centered on the initial empirical evaluation, which was limited to low-dimensional synthetic datasets and prompted concerns regarding practical scalability to higher dimensions. This was addressed by the authors by running additional experiments on the UCI tabular benchmarks. However, I do think that it is worth trying out harder datasets such as ImageNet, as the baseline architectures used in the paper have been applied to them.

On the modeling/theoretical side, I think the authors should provide further discussion on how the Hierarchical Factorized Voronoi restricts the expressiveness of the PC.

Overall, I think the merits of the paper outweigh the weaknesses and would recommend acceptance.